# Permafrost landsystems define regional variability in climate change effects on northern environments

Steven V. Kokelj [1] ✉, Stephen A. Wolfe [2], Niels Weiss[1], Duane Froese[3], Jennifer L. Baltzer [4], Trevor C. Lantz[5], H. Brendan O'Neill[2], Peter D. Morse[2], Anastasia Sniderhan[4], Niek J. Speetjens[5], Jurjen Van der Sluijs[6], Alejandro Alvarez[3], Suzanne E. Tank [7] & Stephan Gruber[8]

Anticipating the environmental and societal consequences of climate-driven permafrost thaw requires knowledge of terrain and subsurface conditions, which prove challenging to obtain at spatial scales necessary for rigorous prediction and decision-making. Analysis of a systematic inventory of permafrost landforms across northwestern Canada demonstrates that landform assemblages co-develop with ecosystems, distinguishing fundamental permafrost properties across a continental-scale ecoclimatic gradient ($10^6$ km²) and among finer-scale ecological regions ($10^3$ to $10^4$ km²). This approach quantifies variation in geological and climatic legacies and delineates the diverse consequences of thaw. Here we show that permafrost landsystems, defined by characteristic landform assemblages, express spatial variation in soil, ground ice, geochemical, and carbon characteristics, enabling these intrinsic conditions to be inferred at regional scales through integrated mapping and analyses. Permafrost landsystems also provide a conceptual framework to inform predictions of thaw-driven change, and to formulate, share, and apply permafrost knowledge across scales, disciplines, and ways of knowing.

Permafrost underlies approximately 12–17 million km² of the land surface in the Northern Hemisphere[1] and profoundly influences physical and biological processes in circumpolar environments. Observed permafrost thaw[2] is projected to increase in the coming century[3]. The development of ground ice gives rise to periglacial landforms, whereas thaw-driven processes that transform ice-rich landscapes produce a wide range of thermokarst landforms, collectively expressing variation in terrain and permafrost conditions[4,5] (Supplementary Fig. 1, Supplementary Table 1). The broad relevance of permafrost to ecosystems, infrastructure, traditional land use, and the global climate[6–8] underscores the need for holistic approaches to characterize permafrost and integrate this knowledge across scales and disciplines to better predict thaw and understand its consequences.

Modeling and remote sensing approaches generate timely, broad-scale spatial information that shapes how the scientific community conceptualizes permafrost thaw[9–12] and how policymakers and the public may perceive it. However, the heterogeneity in permafrost

[1]Northwest Territories Geological Survey, Government of Northwest Territories, Yellowknife, NT, Canada. [2]Geological Survey of Canada, Natural Resources Canada, Ottawa, ON, Canada. [3]Department of Earth and Atmospheric Sciences, University of Alberta, Edmonton, AB, Canada. [4]Department of Biology, Wilfrid Laurier University, Waterloo, ON, Canada. [5]School of Environmental Studies, University of Victoria, Victoria, BC, Canada. [6]Northwest Territories Centre for Geomatics, Government of Northwest Territories, Yellowknife, NT, Canada. [7]Department of Biological Sciences, University of Alberta, Edmonton, AB, Canada. [8]Department of Geography and Environmental Studies, Carleton University, Ottawa, ON, Canada. ✉e-mail: Steve_Kokelj@gov.nt.ca

characteristics and variation in thaw-driven consequences revealed by field research, regional-scale analyses, and the diverse experiences of northerners diverge from the results reported by many of these studies[5,13,14]. Increases in thaw-driven landslides[15,16], ice-wedge melt pond formation[17], thaw-lake expansion, rapid lake drainage[18,19], and peatland degradation[20] are prominent processes that shape thawing terrain over the vast permafrost regions (Supplementary Table 1). The persistent gap between broad-scale predictions and empirical observation impedes the advancement and application of permafrost knowledge across scales and disciplines. Improved conceptual and quantitative understanding of thaw-driven change and ecosystem vulnerability requires knowledge of the terrain and subsurface, particularly ground ice and soil conditions, which is difficult to obtain at regional scales[21,22], where it is necessary for analysis, prediction, and decision-making. However, there is growing recognition that the distribution and dynamics of landforms provide rich information on permafrost properties and on thaw-driven change and ecosystem vulnerability[4,5,23,24] (Figs. 1, 2).

Landsystem approaches have long provided a nested, spatial framework for characterizing terrain patterns that reflect landform-subsurface relations, and, when mapped, they support land-use planning[25,26]. Landsystems also provide insights into past geological processes and climate, such as the nature of glaciation[27]. Within multi-level ecological land classification schemes, landsystems express landform, soil, and vegetation patterns that combine as landform assemblages or "land types"[26] (Supplementary Table 2). Building on traditional landsystem concepts, we propose that geological inheritance and paleoenvironmental history are most intimately linked in the evolution of permafrost landscapes, producing distinct permafrost landform assemblages (Supplementary Table 1) that co-develop with ecosystems to reflect fundamental permafrost properties[10,28–30] (Figs. 1, 3). These landform assemblages, comprised of periglacial landforms – products of cold-climate processes and ground-ice development, and thermokarst landforms resulting from the thaw of ice-rich permafrost, together referred to as permafrost landforms – develop in association with characteristic surficial deposits to organize at regional scales ($10^3$ to $10^4$ km$^2$), forming permafrost landsystems (Figs. 2, 3; Supplementary Table 1).

In this study, we utilize a robust, systematic inventory of permafrost-associated landforms from northwestern Canada[5] to explore variation in the patterns of landform assemblages, and by extension, the range of terrain conditions and thaw sensitivities they reflect (Figs. 1, 2 and Supplementary Table 1; also see Methods). Specifically, we (1) explore the broad-scale patterns of permafrost landform assemblages, and (2) evaluate the influence of climate and terrain conditions on their distribution across a continental-scale biophysical

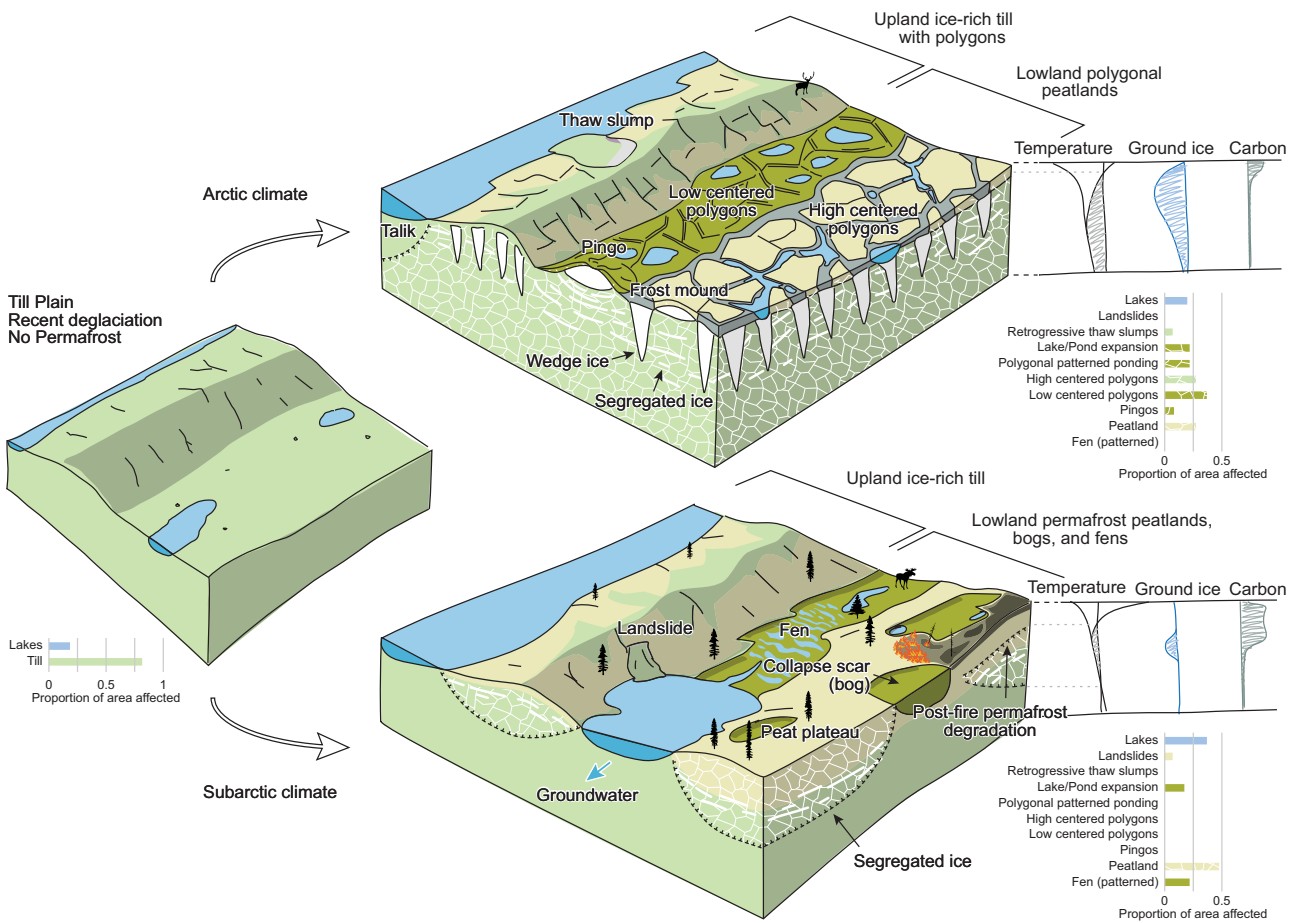

**Fig. 1 | Permafrost landform assemblages are the product of geological inheritance conditioned by climate, geomorphic, and ecosystem factors.** Permafrost landform assemblages, comprising landforms indicating ice-rich terrain or the consequences of thaw, co-develop with ecosystems to characterize a typical range of soil, carbon, ground thermal, and ground ice conditions. Variations in the climate conditioning of similar geological deposits can yield different permafrost landform assemblages distinguished by their surface expressions and subsurface properties. For example, in low-lying terrain in Arctic regions, relatively thin organic deposits host high and low-centred polygons, which are typically rich in wedge and segregated ice. In warmer subarctic climates, thicker organic deposits favor the presence of permafrost, which is absent from adjacent fens and collapse bogs. Thawing permafrost is common here due to the warming climate and is accelerated by fire. Landform assemblages determined through systematic inventories across regions are expressed as unique, universally comparable permafrost fingerprints (bar graphs) that characterize permafrost landsystems.

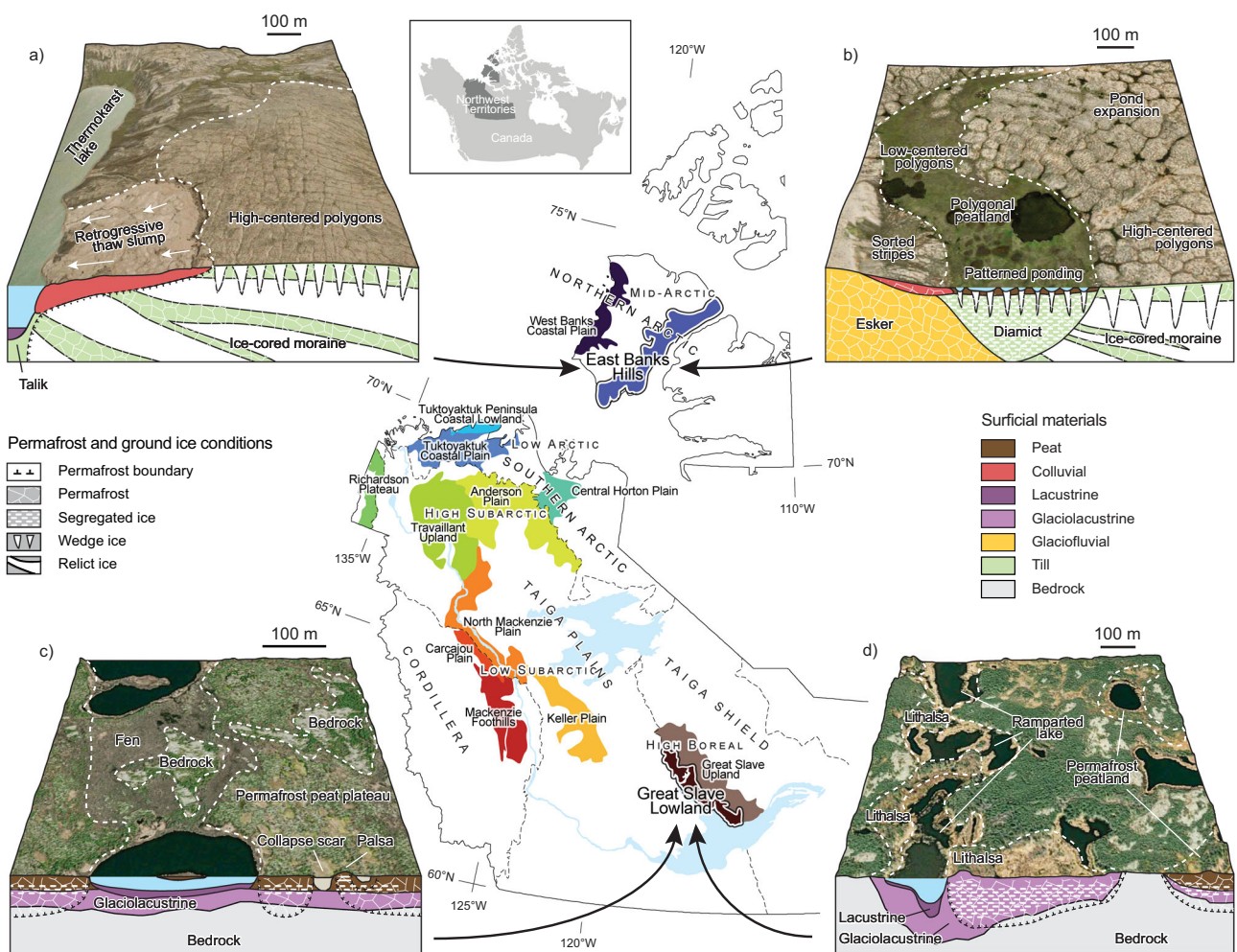

**Fig. 2 | Variation in landscape genesis and climate conditioning of environment components yields permafrost landform assemblages and subsurface properties that differentiate permafrost landsystems.** Permafrost conditions revealed by landforms contrast the East Banks Hills Level IV ecoregion (**a**, **b**) in the Northern Arctic ecozone and the Great Slave Lowland Level IV ecoregion (**c**, **d**) of the Taiga Shield ecozone. **a** Cold continuous permafrost of the glaciated East Banks Hills has preserved relict glacier ice in moraines indicated by retrogressive thaw slumps[30], and Holocene ice wedge development is revealed by high-centered polygons and ponds, which together form dominant, thaw-sensitive landform assemblages. **b** Lowland basins indicate past thaw followed by the development of polygonal peatlands underlain by segregated and wedge ice. Polygonal patterned ponds coalesce into geometric ponds and lakes, forming another typical landform assemblage. **c** In warm discontinuous permafrost of the bedrock-dominated Great Slave Lowland, discrete permafrost peatlands underlain by thin overburden with segregated ice are dissected by permafrost-free bogs and fens. **d** Undulating low hillocks called lithalsas form when segregated ice heaves terrain and typically occur in association with "ramparted-lakes" and ponds affected by shoreline thermokarst, forming a thaw-sensitive permafrost landform assemblage in an otherwise thaw-stable landscape[41]. The Inset map of the Northwest Territories, Canada, spanning 15 degrees of latitude, shows major ecozones (Level II ecoregions), nested ecoclimate regions (Level III ecoregions), and highlighted "finer-scale" Level IV ecoregions considered in this study (Supplementary Table 2). High-resolution base imagery for (**a**–**d**) retrieved from the ESRI World Imagery layer (accessed January 2024). Map adapted from Level IV ecoregions[31–35]. Ecoregion outlines and other spatial layers derived from https://www.apps.geomatics.gov.nt.ca/arcgis/rest/services.

gradient ($10^5$ km²), (3) determine the influence of geologic, and paleoenvironmental history on variation in landform assemblages and permafrost conditions within and among smaller, landsystem-scale ecoregions ($10^3$–$10^4$ km²) (Figs. 2, 3), and (4) demonstrate that permafrost landsystems offer a conceptual framework for understanding the spatial variation in permafrost properties and thaw trajectories from quantitative analysis of permafrost landforms. An ecological classification for the Northwest Territories (NWT), Canada, defines the spatial extent of our study and structures the analyses[31–35] (Fig. 2 and Supplementary Table 2). A subsample of the NWT-wide permafrost landform inventory used in this study captures a range of permafrost landforms that reflect major ground-ice types, as well as soil, ground temperature, and ecological conditions encountered across northwestern Canada[5,10] (Supplementary Fig. 1 and Table 1). These data are from a 7.5 × 7.5 km grid-based inventory of the presence or absence of

28 permafrost landform types (Supplementary Table 1) across 3292 grid cells. The observed landforms within a grid cell combine to form assemblages and reflect permafrost conditions (Supplementary Table 1). Feature identification is constrained by the use of Sentinel-2 imagery with a 10 m pixel size, which was available for the entire study region at the initiation of the NWT Thermokarst Collective mapping project[5]. The sampled grid cells represent the total areas of 14 Level IV ecoregions nested within seven ecozones[5] (Level II ecoregions) (Fig. 2; Supplementary Table 2). Redundancy analysis (RDA) was applied to a subsample of the grid-based inventory of permafrost landform types (1,959) across the Level IV ecoregions (Supplementary Table 2) to examine general associations between landform assemblages and primary climate and terrain conditions (Supplementary Table 3). To visualize broad-scale patterns in the RDA site scores, grid cells (sites) are labeled by ecozone ($10^5$ km²) (Fig. 2 and Supplementary Table 2).

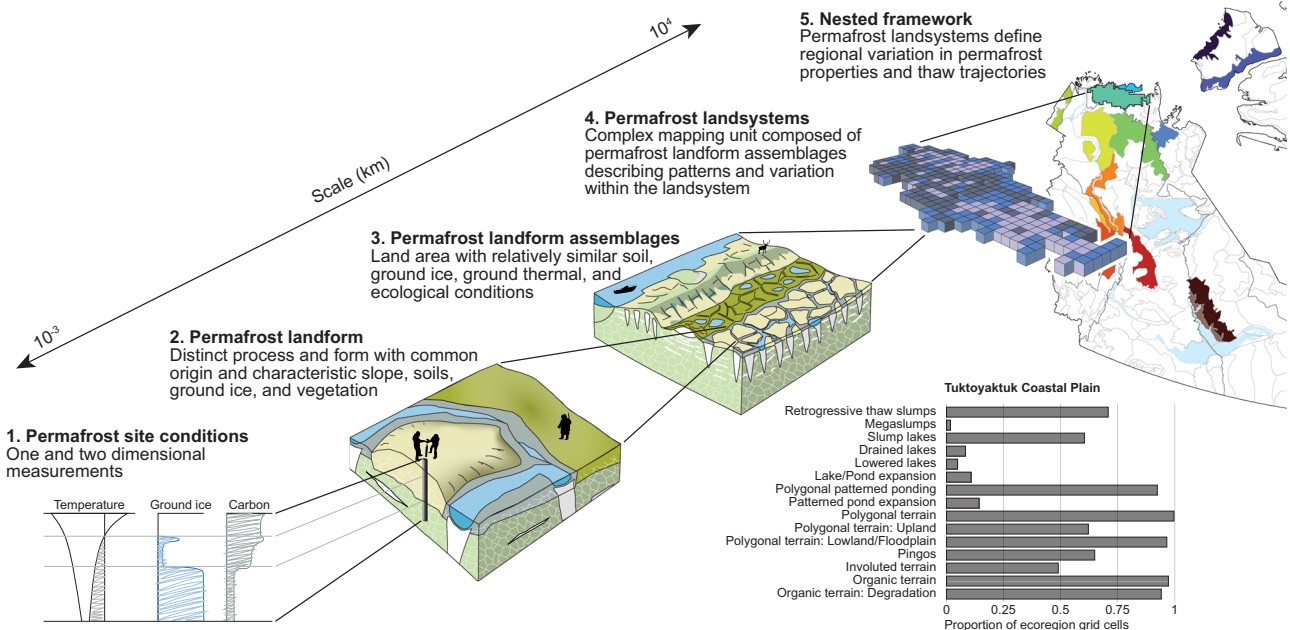

**Fig. 3 | Permafrost landsystems provide a framework enabling the transfer of knowledge of site-specific permafrost conditions across spatial scales by mapping landform assemblages.** Permafrost landforms associated with specific ranges of soil, ground ice, and ground temperature conditions combine to form assemblages that have co-developed with ecosystems, reflecting thaw sensitivities and ecosystem vulnerabilities (1–3). Mapping or inventorying landform assemblages allows spatial patterns to be assessed and variation in soil, ground ice, and thermal conditions to be inferred within and between larger areas at the scale of a regional landsystem (3–4). A permafrost landsystem contains combinations of recurring landform assemblages (4). The bar graph describes the composition of assemblages within a landsystem, allowing regional variation in permafrost conditions to be understood and facilitating comparisons between landsystems. Integrated and systematic permafrost landform mapping is an essential step towards better understanding the linkages between landforms, permafrost conditions, and ecosystems, as well as how these conditions vary within and between landsystems (regional scales). The coupled mapping and conceptual framework provide an appropriate context for upscaling and communicating study results, supporting a better appraisal of the environmental and social consequences of climate change and thawing permafrost. Ecoregion outlines and other spatial layers derived from https://www.apps.geomatics.gov.nt.ca/arcgis/rest/services.

Fourteen Level IV ecoregions provide spatial boundaries for exploring finer-scale regional patterns of variation in permafrost landform assemblages as they are conceptually similar to landsystems, representing regions ($10^3$–$10^4$ km$^2$) of recurring landform, soil, and vegetation conditions[26,31–35] (Fig. 2 and Supplementary Table 1 and Appendix 1). For landsystem-scale analyses, the RDA scores are stratified by Level IV ecoregions. The richness of permafrost landforms across ecoregions was compared by creating landform accumulation curves from inventory data for 3292 grid cells within Level IV ecoregions. We also developed a clustered heatmap to explore the natural groupings of permafrost landform assemblages and how these influence similarities and clustering of Level IV ecoregions. The data used in these analyses are available in Weiss et al.[36].

This study demonstrates that combinations of permafrost landforms are linked to variation in permafrost conditions across a continental-scale ecoclimate gradient and among landsystem-scale ecological regions (Figs. 2, 3 and Supplementary Table 2). Here we show that a permafrost landsystem approach enables spatial and stratigraphic variation in soil, carbon, ground ice, and thermal conditions to be inferred at regional scales based on the distribution of mapped permafrost landform assemblages (Figs. 1, 2) and provides a conceptual framework for integrating knowledge of permafrost properties, ecosystems, and trajectories of thaw across sites, spatial scales, and disciplines (Fig. 3).

## Results
### Landform assemblages reflect variation in permafrost characteristics across continental-scale gradients
Permafrost landforms across northwestern Canada exhibit patterns associated with climate and terrain variables (Fig. 4). The RDA was statistically significant, explaining 23.3% of the variation in the composition of permafrost landforms within the inventoried grid cells ($F_8 = 74.27$, $p = 0.001$). All environmental variables were significant ($p < 0.01$). The analysis explained 16.6% of the variation across axis 1 (RDA1, $F_1 = 420.82$, $p = 0.001$) and 3.2% across axis 2 (RDA2, $F_1 = 80.26$, $p = 0.001$). Axis 1 separates data along a continental-scale climate gradient, and axis 2 distinguishes site scores in relation to topographic relief.

Permafrost landform assemblages capture a continental-scale gradient in permafrost conditions, and reflect significant variation in substrate, ground ice, and ground temperature conditions that characterize the heterogeneity of this broad ecoclimate transition indicated by the site scores along RDA axis 1 (Fig. 4a). Permafrost landform assemblages in lowlands of the Taiga Plains, Low Subarctic and Taiga Shield, High Boreal (negative RDA1 scores) are associated with the highest mean annual air temperatures, rainfall, and forest fire burn area (Fig. 4). These conditions are associated with permafrost peatland-dominated landform assemblages dissected by dendritic and multi-basin drainage patterns with permafrost-free string bogs, collapse scars, and thaw ponds, and thaw-lake complexes (ramparted-lake lithalsas) which together reflect thin and patchy permafrost (Fig. 2c, d). Sites that cluster across the origin of RDA1 reflect diverse lowland landforms that characterize the range of ground temperatures and ground ice associated with variation in permafrost conditions encountered across the Taiga Plains, High Subarctic, of which the northern extent approaches the forest-shrub transition zone, and colder ground temperatures. The Southern and Northern Arctic ecozones plot positively on RDA axis 1, with sites in the lower-right quadrant reflecting strong associations between lowland landforms associated with wedge ice and related hydrological features (Fig. 2a, b).

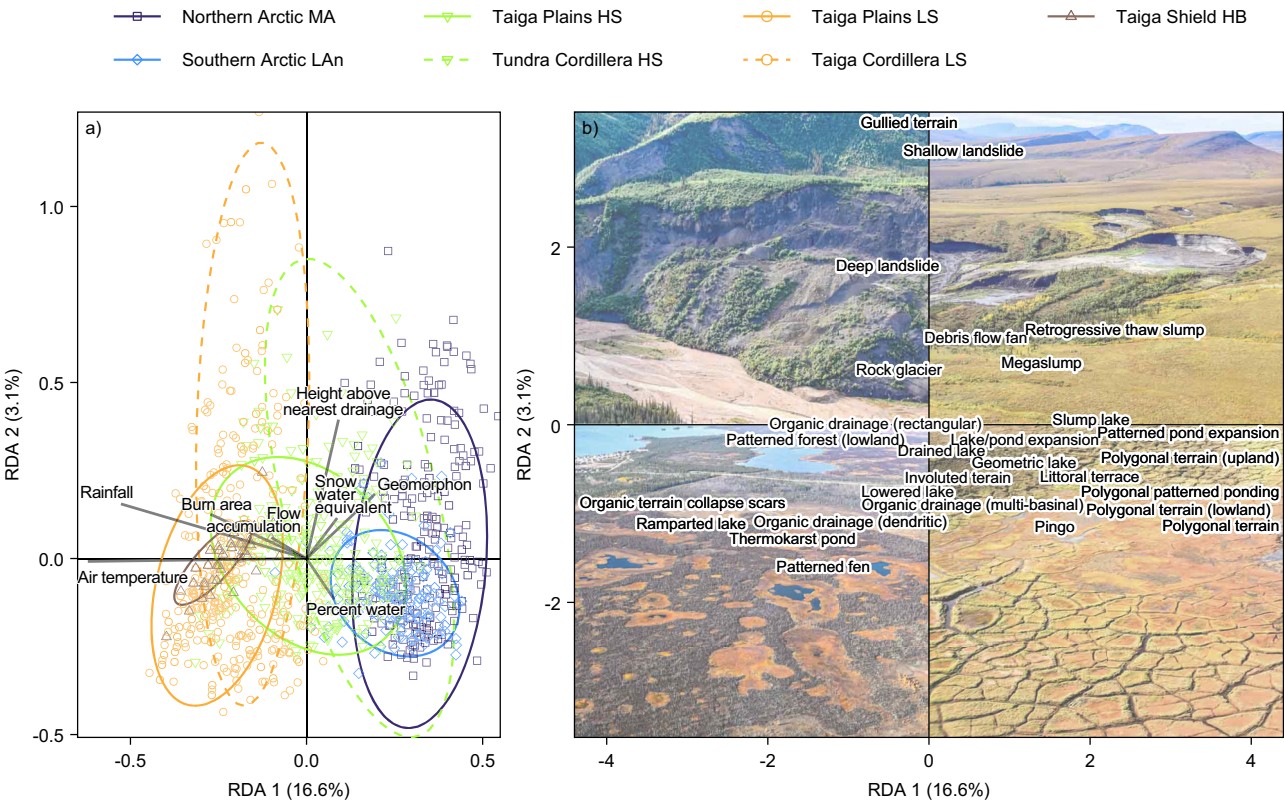

**Fig. 4 | Permafrost landform assemblages quantify variation in permafrost properties across broad spatial scales.** Redundancy analysis results showing (**a**) site scores reflecting landform assemblages, stratified by ecozones and vectors indicating loadings of the RDA1 and RDA2 environmental predictors, and (**b**) the loading scores of the 28 permafrost landforms used in the RDA. RDA1 separates site scores along a continental-scale climate gradient, and RDA2 separates scores along a gradient of topographic complexity. The Level II ecozones are subdivided by Level III ecoclimate regions shown in Supplementary Table 2. As expressed in the legend, the Taiga Plains and Cordillera are subdivided into High and Low subarctic (HS, LS), and the Northern Arctic into the Mid-Arctic (MA), and the Southern Arctic into the Low Arctic north (LAn), and the Taiga Shield into the High Boreal (HB).

Supplementary Tables 1 and 3 provide information on permafrost landforms and primary environmental variables used in the RDA. Oblique photographic images provided using Open Government License from the Northwest Territories Thermokarst Mapping Collective, Government of Northwest Territories, Northwest Territories Geological Survey; https://www.apps.geomatics.gov.nt.ca/arcgis/rest/services/GNWT_Operational/NTGS_ThermokarstCollective_Operational/MapServer and the Ecological Land Classification Photo Inventory Government of Northwest Territories, ECC, Fire Management Division; https://www.apps.geomatics.gov.nt.ca/arcgis/rest/services/GNWT/BiologicEcologic_LCC/MapServer/3).

These landform assemblages occur with low MAAT and rainfall, reflecting contemporary permafrost conditions and the legacy of a cold Holocene climate (Fig. 4).

Positive RDA2 scores express the association between mass wasting features, greater height above nearest drainage, and variation in terrain position classes (geomorphons) (Fig. 4a), characteristic of foothills or plateaus in the transition between the Taiga Plains and Cordillera, or localized areas of greater topographic relief in the Taiga or Arctic ecozones. Topographically complex environments in discontinuous permafrost (upper left quadrant of the RDA) are associated with shallow landslides and large slides that fail at the base of thin permafrost[37]. Terrain with retrogressive thaw slumps plot in the upper right quadrant of the RDA (Fig. 4b) and reflects geomorphically complex (RDA2) ice-rich landscapes in cold permafrost (RDA1) of the Northern Arctic ecozone. Across the climate gradient expressed by RDA1, negative RDA2 scores are associated with low topographic variability and reflect poorly drained, water-rich landscapes with string bogs and dendritically drained permafrost peatlands in discontinuous permafrost of subarctic and boreal regions, and lowland polygonal terrain in Arctic areas.

Over 75% of the variation in the composition of permafrost landforms across the inventoried grid cells remains unexplained by the environmental variables used in the RDA, indicating the limitations of the common practice of applying broad-scale spatial data without field-informed evidence in modeling permafrost conditions[10,38–40].

Improved predictor-variable datasets may reduce unexplained variation. However, additional analysis, including derived and highly correlated variables such as surficial geology, modeled ground ice abundance, and permafrost probability (Supplementary Table 3), did not improve the explanatory power of the RDA. The high percentage of unexplained variation indicates that variation in geology, climate, and ecological conditions needs to be better accounted for in spatial analyses, and that systematic landform mapping has significant potential to advance quantitative understanding of variability in permafrost terrain conditions, intrinsic properties, and thaw-driven change trajectories[5,23].

### Landforms reveal variations in permafrost conditions between landsystems

The patterns of variation in permafrost landform assemblages distinguish landsystems and provide insights into regional-scale differences in permafrost conditions and thaw trajectories. Figure 5 shows RDA scores and landform richness grouped by Level IV ecoregions, and Fig. 6 is a clustered heatmap that explores the natural groupings of permafrost landforms into assemblages, and how these influence similarities and clustering across the landsystem-scale Level IV ecoregions. Coupled with these analyses, we also apply permafrost landsystems as a conceptual framework to guide narratives that explain how geological legacies, physiography, and climate have shaped spatial and compositional variation in landform assemblages, intrinsic

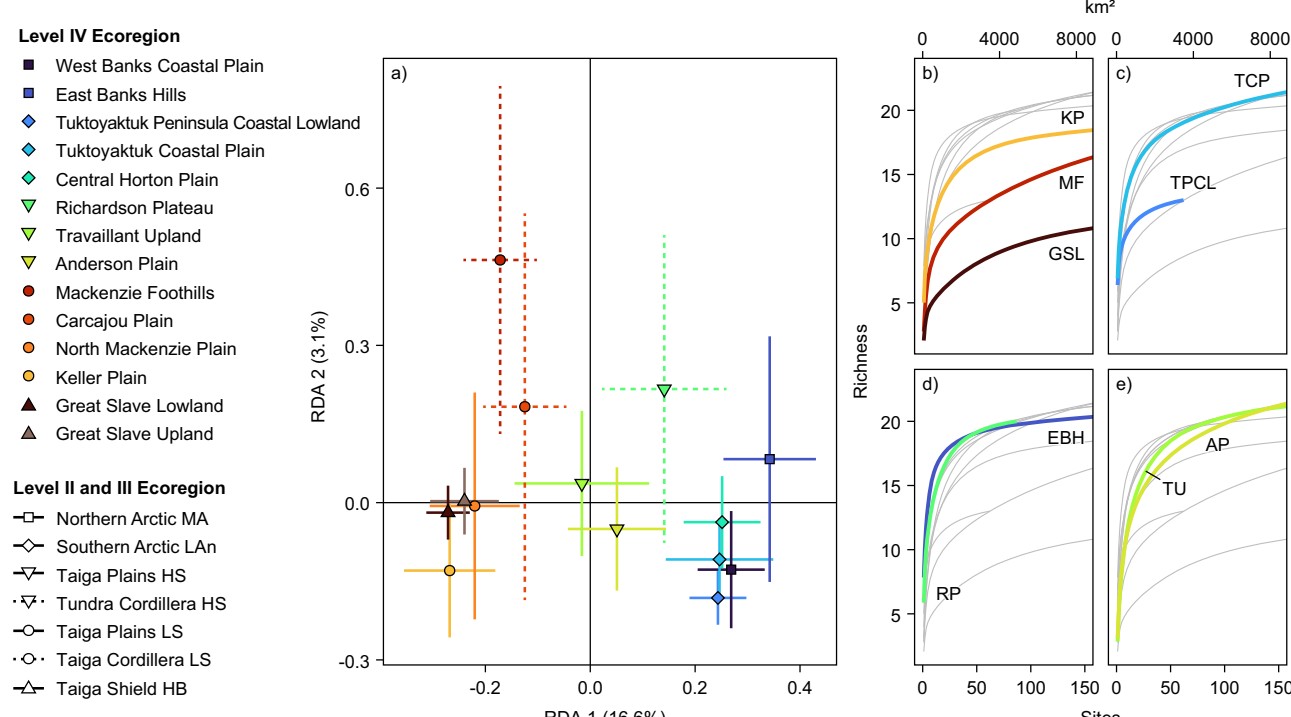

**Fig. 5 | Variation and richness in landform assemblages summarized by different Level IV ecoregions or landsystems. a** Redundancy analysis results of landform assemblages grouped by Level IV ecoregions showing RDA1 and RDA2 centroids and standard deviation of site scores. **b–e** Landform richness as a function of mapping effort measured by the total number and area of the grid cells. **b** Level IV ecoregions with different geological conditions within the Taiga Cordillera (Mackenzie Foothills, MF), Taiga Plains (Keller Plain, KP), and Taiga Shield (Great Slave Lowland, GSL) ecozones; and (**c**) for glaciated (Tuktoyaktuk Coastal Plain, TCP) and unglaciated terrain (Tuktoyaktuk Peninsula Coastal Lowland,

TPCL), within the Southern Arctic ecozone. **d** Level IV ecoregions with similar glacial legacy but contrasting climate and physiography (East Banks Hills, EBH, - Northern Arctic; Richardson Plateau, RP - Tundra Cordillera). **e** Level IV ecoregions within ecotones (Travaillant Uplands, TU, & Anderson Plain, AP) in the Taiga Plains. The light gray lines on (**b–e**) show richness curves for all Level IV ecoregions. Supplementary Appendix 1 shows satellite imagery and oblique aerial photographs for each Level IV ecoregion in (**b–e**). In (**a**), the point shape indicates the ecozone (Level II), with an abbreviation for ecoclimate region (Level III) (Supplementary Table 2).

permafrost properties, and thaw trajectories (Figs. 2, 5 and Supplementary Appendix 1). From these quantitative analyses, we draw five main insights:

(1) *Geological boundaries demarcate significant differences in permafrost landform assemblages.* In the discontinuous permafrost zone, the narrowest RDA confidence intervals and lowest landform richness and terrain coverage characterize ecoregions (i.e., Great Slave Lowland) within the bedrock-dominated Taiga Shield (Figs. 5a, b and 6). Ice-rich permafrost is confined to pockets of fine-grained sediments, manifesting as ice-rich lithalsa, thaw-lake assemblages, or discrete permafrost peatlands (Figs. 2c, d and 6), reflecting distinct trajectories of thaw-driven change[41] in an otherwise thaw-stable Taiga Shield landscape. In contrast, the greater landform richness and RDA1 standard deviation (STDev) of the Keller Plain are associated with extensive, thawing peatlands and shallow thermokarst lakes (Figs. 5a, b and 6) reflecting warm and thin permafrost of the organic-rich, glacially-scoured Taiga Plains, Low Subarctic (Fig. 6). The Mackenzie Foothills of the Taiga Cordillera (Fig. 2), is characterized by a gradually increasing landform richness curve that indicates high between-site diversity and relatively low overall richness: a product of geologic and physiographic heterogeneity, and large areas of exposed bedrock or regolith. The variable physiography and substrate yield a diversity of landslide types along incised valleys, degrading permafrost peatlands on till plains, and thaw-lake complexes in glacial meltwater channels, comprising distinct landform assemblages (Fig. 6). Within the Taiga biome, the dendrogram clusters bedrock-dominated landsystems distal to the organic-rich Keller Plain (Fig. 6).

(2) *Within the same ecoclimate region (Low Arctic north), glaciated and unglaciated terrain hosts distinct landform assemblages.* Widespread wedge ice comprises dominant lowland landform assemblages reflecting cold, continuous permafrost across these tundra landsystems[29] (Figs. 1, 5a, c and 6). However, the complexity of glacial deposits, variation in post-glacial terrain evolution, and patchy preservation of relict massive ice[42] yield greater landform richness and between-site diversity in the *glaciated* Tuktoyaktuk Coastal Plain than in the adjacent *unglaciated* Coastal Lowland, a sandy outwash plain with shallow, oriented lakes distributed throughout[43] (Fig. 5a, c). On the Tuktoyaktuk Coastal Plain, ice wedges in uplands and lowland organic terrain, thaw slumps, pingos, and involuted terrain comprise landform assemblages reflecting diversity in terrain and ground ice conditions, and hence, thaw-trajectories[18,29,30,42], contrasting widespread polygonal patterned ponding and partial drainage of oriented lakes on the Coastal Lowland (Fig. 6).

(3) *Holocene climate drives differences in permafrost landform assemblages between ecoregions with similar geological legacies.* Permafrost-preserved ice-cored moraines share common landforms, such as thaw slumps, but variations in past climate and physiography yield differences (Figs. 5a, d and 6). The richness curve for East Banks Hills in the Northern Arctic ecozone rises rapidly and then flattens, indicating a diversity of landforms throughout the landsystem (Figs. 5d and 6). Preservation of relict glacier ice and extensive development of ice-wedge networks, indicated by abundant RTS[15,30] and polygonal pond expansion[17], comprise thaw-sensitive upland assemblages reflecting glacial legacy and a cold Holocene climate (Figs. 2a,

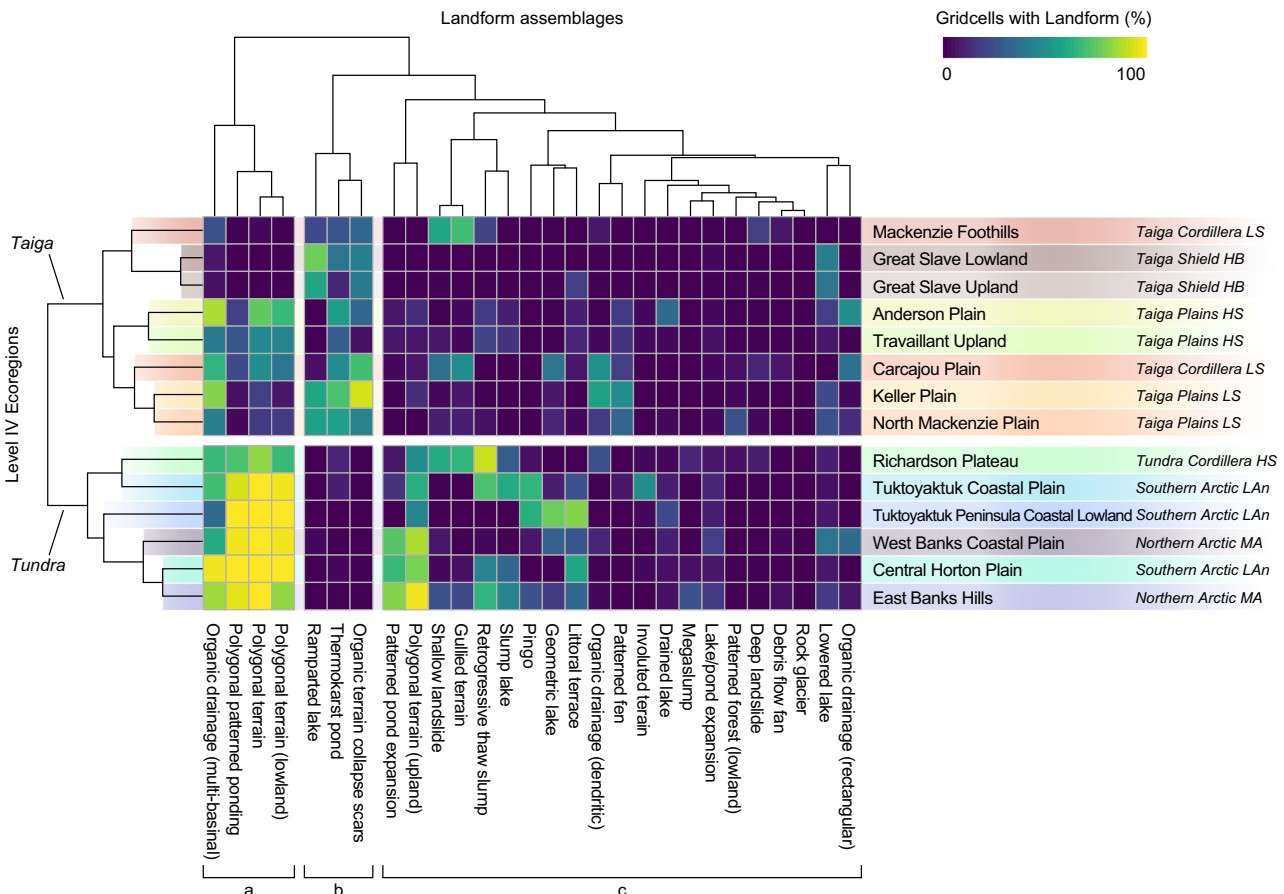

**Fig. 6 | Clustered heatmap illustrating the percentage of grid cells characterized by different permafrost landforms within Level IV ecoregions.** The heatmap illustrates how permafrost landforms (horizontal axis) cluster as assemblages for the 14 a priori Level IV ecoregions (vertical axis). The Ecoregion labels indicate Level IV ecoregions and the Level II & III ecoregions they nest within are in italics. The landform dendrograms hierarchically cluster variation, expressing common permafrost landform assemblages. The Level IV ecoregion dendrogram displays their similarity and hierarchical clustering based on landform assemblages, and the first-order split divides the Tundra and Taiga biomes. In the landform assemblage dendrogram, the first-order split separates (**a**) lowland polygonal peatland assemblages from the remainder of landforms. The second-order split separates (**b**) ramparted thermokarst lake and pond, and peatland collapse scar assemblages, predominantly in the Taiga, from (**c**) the remainder of the landforms. Supplementary Table 1 provides information on permafrost landforms. Supplementary Table 2 provides information on the Northwest Territories Ecological Land Classification framework.

b and 6). The larger RDA STDev and a slightly gentler rising richness curve for the Richardson Plateau of the Tundra Cordillera reflect its ecotonal nature and greater topography. Peatlands with dendritic drainage patterns transition northwards to polygonal terrain. RTS and shallow slides are common, along with a few megaslumps in the fluvially incised plateau and gullied terrain, and deep-seated failures occur on steeper slopes underlain by sedimentary bedrock[44]. Preservation of relict glacier ice produces similar conditions across ice-cored moraine landsystems; however, the greater variation in past and present climate and topography in the Richardson Plateau yields diverse landform assemblages and thaw-driven changes (Fig. 5a, d). The dendrogram represents these two regions of similar glacial legacy and contrasting climate at opposite branches of the landsystems we sampled within the Tundra biome (Northern Arctic, Southern Arctic, Tundra Cordillera) (Fig. 6). However, landform assemblages of the Richardson Plateau are most similar to the nearby glaciated Tuktoyaktuk Coastal Plain of the Southern Arctic, reflecting terrain of common glacial origins and post-glacial climate (Fig. 6).

(4) *Landsystems within ecotones express the greatest diversity in landforms and permafrost conditions.* Within the northern part of the Taiga at the boundary with the Tundra biome (Fig. 2), the Travaillant Upland and Anderson Plain have the highest RDA1 STDev and landform richnesses (Fig. 5a, e). High diversity within these landsystem-scale

Level IV ecoregions is reflected by continually rising richness curves (Fig. 5e). Thawing peatlands, patterned fens, and thermokarst ponds comprise landform assemblages that transition northwards to polygonal peatlands and ponds. Variations in geology and preservation of ground ice yield thaw-sensitive slopes in areas with greater topographic relief (Figs. 1, 6). A wide range of landform assemblages indicates diversity in permafrost conditions and, hence, thaw trajectories within Level IV ecoregions situated within a major ecotone, distinguishing these High Subarctic Taiga landsystems from those within the Low Subarctic to the south (Fig. 6).

(5) *Landscapes in discontinuous permafrost or with few excess ground-ice types have lower landform richness.* Three of the four ecoregions with the lowest landform diversity are underlain by discontinuous permafrost, and with bedrock areas, and where segregated ice is the only common excess ground ice (Fig. 5b). In continuous permafrost, landform homogeneity and low landform richness of the unglaciated Tuktoyaktuk Peninsula Coastal Lowland characterizes a sandy outwash plain that only includes appreciable ground ice in the form of ice wedges[43] (Fig. 5c). As reflected in the results of the RDA, landform richness curves, and a clustered heatmap (Figs. 5, 6), these regions contain thaw-sensitive permafrost terrain, but the nature of thaw-driven changes and environmental consequences is less variable than in other permafrost environments, where ground-ice types are more diverse and abundance is greater.

## Discussion

Here we demonstrate that periglacial landforms, which are the products of cold climate processes and ground-ice development, and thermokarst landforms manifesting the consequences of thaw[5], comprise landform assemblages that reveal spatial variation in fundamental permafrost properties and distinguish differences between permafrost landsystems ($10^3$–$10^4$ km$^2$) (Figs. 5, 6). Systematic landform inventories over large areas and analyses to determine the composition of typical assemblages can overcome regional-scale data sparseness in permafrost science. These mapping methods quantify spatial patterns of variation that reveal subsurface conditions, inherently linked to trajectories of landscape change, ecosystem vulnerability, and permafrost carbon feedbacks (Figs. 1, 2). Permafrost landsystems also provide a conceptual framework for integrating knowledge of site-level permafrost conditions into landform assemblages and mapping them at the regional scale (Fig. 3). Thus, adopting a permafrost landsystem framework enables the integration of knowledge critical to inform landuse, community, and infrastructure planning, guide environmental assessments, and upscale research results with appropriate context.

Analyses of systematically mapped landforms from northwestern Canada reveal that a wide range of assemblages reflect the diversity of permafrost conditions across broad ecological regions ($10^5$–$10^6$ km$^2$), and organize to distinguish regional-scale landsystems ($10^3$–$10^4$ km$^2$) (Figs. 4–6). By extension, this approach discerns variation in permafrost properties reflecting different geological legacies, physiography, and climate. For example, dominant lowland assemblages transition from permafrost peat plateaus with collapse scars and fens in the Taiga, to polygonal peatlands, patterned ponds, and pingos in the Tundra (Figs. 4–6). High variability in landforms characterizes the ecotonal landsystems at the boundary between Taiga and Tundra biomes (Figs. 4–6). Multi-landform mapping enables quantitative exploration of within- and between-region patterns by using techniques commonly used in the ecological sciences, providing insights into variation in regional-scale permafrost conditions and thaw trajectories (Figs. 5, 6). Though the RDA clearly discerns these patterns, most of the landform variation is unaccounted for by available environmental and climate predictors, highlighting the limitations of their frequent use in broad-scale spatial modeling of permafrost conditions[10,38–40], without adequate field-based information. Improved predictor-variable datasets may reduce unexplained variation. However, additional analysis, including derived and highly correlated variables such as surficial geology, modeled ground ice abundance, and permafrost probability (Supplementary Table 3), did not improve the explanatory power of the RDA. The unexplained variation can be attributed to the greater heterogeneity in permafrost conditions captured by landform assemblages than can be resolved with the available predictor datasets. This underscores a critical need for an integrated landform mapping approach, further strengthened by fine-scale surficial geology and soil mapping and topographic data, to advance understanding of permafrost conditions and the consequences of thaw within and between permafrost regions across the vast and heterogeneous northern landscape.

Landform mapping provides insights into the spatial variation of permafrost conditions[4,5,23,24]. Global examples demonstrate the use of landforms to reconstruct past permafrost environments[45], estimate modern permafrost extent[46] or ground-ice patterns[29,47], assess geohazards[16,37], and parameterize and test spatial models[48,49]. Mapped landforms, inherently linked to ground ice and ecosystems[13,29,50], provide spatial evidence of variation in permafrost conditions, extending what is offered by heuristic approaches to estimate ground ice content[10] or assess the vulnerability of permafrost carbon[11,38]. Indeed, variation in the composition of landform assemblages provides rich, spatially explicit insights into climate change vulnerabilities and the ecological, biogeochemical, and societal consequences of thaw, linking local to regional spatial scales (Figs. 1, 3). Thus, the study of permafrost geomorphology, landform evolution, and biophysical feedbacks[41,43,51–53] provides critical knowledge of the soil, carbon, thermal, and ground ice properties of landforms and assemblages (Fig. 3) necessary to extrapolate the trajectories and consequences of thaw within the mapping framework of permafrost landsystems.

Here, we explored the composition of permafrost landform assemblages stratified by ecozones and finer, landsystem-scale ecoregions to demonstrate that systematic mapping information can reveal variation in permafrost properties and consequences of thaw across spatial scales (Figs. 4–6). Applying hierarchical clustering to determine the composition of permafrost landform assemblages and their organization into landsystems provides a quantitative basis for evaluating variation in permafrost conditions within and between regions (Fig. 6). Critically, however, it also offers tremendous potential to extrapolate site-specific knowledge to other permafrost areas with similar landform assemblages. Mapped landform inventories can provide training datasets for supervised classification and spatial modeling to predict landform distributions[49,54], with the potential for modeled outputs to reflect information on permafrost configuration, soil properties, carbon storage, and ground ice conditions. Integrating the mapping of landform assemblages and landsystems, supported by a conceptual framework linking terrain and subsurface conditions, into numerical modeling activities can provide appropriate context across multiple scales for designing, parameterizing, and interpreting transient simulations that explore the magnitude, timing, and consequences of thaw. Thus, identifying spatial variation in permafrost landform assemblages may inform model parameterization for specific locations[55] or subgrid schemes employed in grid-based models[56,57], with the growing potential to laterally couple representations of commonly associated and key assemblages[58].

In conclusion, our study demonstrates that permafrost landform assemblages and regional-scale permafrost landsystems, derived through integrated mapping and multivariate statistical techniques, enable variation in permafrost conditions and thaw trajectories to be better understood. Based on variation in soil, ground ice, ecosystems, and landform distributions, the permafrost landsystems framework provides a mapping and analytical approach to build knowledge of permafrost environments across spatial scales, offering the potential for significantly advancing regional-scale understanding of variability in permafrost conditions (Fig. 3). Permafrost landsystems also provides a conceptual framework to view landscapes as entities of inter-related terrain and ecosystem components linked through geological legacy and climate history. Thus, the co-development of permafrost landforms and ecosystems necessarily informs forward-looking trajectories of change. The permafrost landsystems framework also provides a foundation for communicating ideas and developing concepts across disciplines and between knowledge systems because it fosters linkages among site and regional conditions, theory and simulation, and knowledge holders' observations and experiences (Figs. 1–3). The permafrost landsystem framework enables knowledge transfer from site to region with appropriate context, thereby better connecting permafrost field conditions and management issues with modeling approaches, synthesis products, land-use planning, and policy development. Permafrost landsystems provide a critical analytical and conceptual framework for formulating, sharing, and applying permafrost knowledge across scales, disciplines, and ways of knowing (Fig. 3), essential to better understanding the interconnected and diverse effects of climate-driven permafrost thaw locally and globally.

## Methods

### Study area and permafrost landform inventory

Permafrost landforms indicating thaw-sensitive terrain are being systematically inventoried for the entire Northwest Territories, Canada[5], covering an area of more than 1.3 million km$^2$. In this study, we sampled a total of 14 Level IV ecoregions comprising a 114, 187 km$^2$ area that

span several ecozones from discontinuous permafrost in the Taiga Plains to cold continuous permafrost in the Arctic, bounded by bedrock-dominated Taiga Shield to the east and the Taiga and Tundra Cordilleran to the west[59] (Fig. 2). Most of these regions are composed of glaciated terrain underlain by permafrost with a wide range of ground ice[10] and ground thermal conditions[2]. Variations in geology and climate yield biophysical diversity, which is broadly grouped into the Taiga Plains, Taiga Shield, Taiga Cordillera, Tundra Cordillera, and the Northern and Southern Arctic ecozones[31–35] (Supplementary Table 2). This multi-scale, nested framework delineates ecological regions based on climate, physiography, and vegetation[31], which we apply in this study to guide sampling design and to explore the permafrost landform data. The inventory of 28 landform types characterizes thermokarst landforms that develop due to the thaw of ice-rich permafrost, and periglacial landforms that reflect intrinsic terrain properties, which together are the product of geological legacy, and the interaction of cold region geomorphic processes and biophysical factors, conditioned by past and present climate conditions (Figs. 1, 2, Supplementary Fig. 1 and Supplementary Table 1). Systematic observations of these landforms, evaluated within a $7.5 \times 7.5$ km grid cell framework, described in Kokelj et al. [5] and references within, were made from 14 Level IV ecoregions ($10^3$–$10^4$ km$^2$) across the six ecozones to capture the range of climate, ecosystem, and geological conditions across the Northwest Territories (Fig. 2 and Supplementary Appendix 1). Sentinel-2 imagery (2016–2017; 10 m resolution), viewed at a scale of 1:20,000 was used for this inventory. Level IV ecoregions, mapped at a scale of 1:500,000, are conceptually similar to landsystems because they describe broad recurring vegetation and landform patterns[31] (Supplementary Table 2). The presence or absence of inventoried features comprises a subsample of 3292 mapped grid cells or "sites" describing permafrost landform distribution over a 114, 187 km$^2$ area, which is a subset of the 1.3 million km$^2$ domain of the NWT Thermokarst Mapping Collective[5]. Grid cells with water body coverage greater than 60% were excluded from the analysis.

## Statistical approach

Redundancy Analysis (RDA) was employed to quantify how the presence or absence of permafrost landforms is constrained by primary environmental variables and how they assemble across a latitudinal gradient of ecozones. The multivariate ordination explores the associations between a response matrix of variables, specifically the presence or absence of permafrost landforms (Supplementary Table 1), and a matrix of independent environmental variables on a 1959 grid cell subset of the total 3292 grid cells across the 14 Level IV ecoregions (Supplementary Tables 2 and 3). We used primary environmental variables commonly associated with variation in permafrost and geomorphic conditions. The eight variables retained (Supplementary Table 3) included thirty year means (1991–2020) for: 1) annual air temperature (Air temperature), 2) total annual rainfall (Rainfall), and 3) maximum snow water equivalent, as well as 4) geomorphons, 5) mean height above nearest drainage, 6) flow accumulation, 7) percentage of water area (Percent water), and 8) percentage of burn area for 1991–2020 (Burn area). The final analysis did not use derived and highly generalized (surficial geology) or modeled (ground ice and permafrost probability) categorical variables[10,60] (Supplementary Table 3). Testing revealed the high collinearity between the primary and derived variables, and their addition improved the total explained variation of the RDA by less than 2%.

Statistical analyses were performed in R version 4.5.1[61], using functions in the *vegan* package[62]. The RDA was performed using the rda function, with scaling applied to the environmental variables. We applied variance inflation factors to assess collinearity in the RDA dataset using *vif.cca* and removed highly collinear environmental or climate variables from the analysis. The combined contribution of each

landform feature on RDA1 and RDA2 was calculated as:

$$c = \sqrt{(a^2 + b^2)}$$

where a is the landform score on RDA1 and b is the landform score on RDA2 to calculate contribution c. To compare variation in RDA scores between Level IV ecoregions we plot their RDA1 and RDA2 centroids and standard deviations.

We created landform accumulation curves using landform inventory data from 3292 mapped grid cells to compare the richness of permafrost landforms among first-order landsystems (Level-IV ecoregions) using the *accumcomp* function from the BiodiversityR package[63], using "exact" method. Here, sampling effort refers to the number of grid cells mapped in the ecoregion, and landform richness is the number of permafrost feature types observed. The heatmap with dendrograms in Fig. 6 was created using the *pheatmap* package[64]. Data from 3292 grid cells were summarized as the proportion of gridcells with a landform present within each Level IV ecoregion. The clustering method was complete-linkage clustering using Euclidean distances and without scaling. The datasets used in these statistical analyses are available in Weiss et al.[36]. Final graphics in Figs. 4–6 were optimized using Adobe Illustrator[65]. Hand drawn schematics, mapping, and data graphic elements integrated to create Figs. 1–3 and Supplementary Fig. 1 were also developed in Adobe Illustrator[65].

## Data availability

The permafrost landform and environmental predictor datasets analysed in this study are available at Figshare https://doi.org/10.6084/m9.figshare.31095160. Additional information and Source Data files for Figures are provided with the Supplementary materials. Source data are provided with this paper.

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

## Acknowledgements
This research was supported by the NWT Cumulative Impact Monitoring Program (S.V.K.: 164), the Climate Change Action Plan Implementation of the Government of Northwest Territories, the Climate Change Geoscience Program (Geological Survey of Canada), the Polar Continental Shelf Project of Natural Resources Canada (S.V.K.: 303-25, 309-24, 317-23, 321-22, 303-21; D.F.: 630-22, 681-23, 669-24, 665-25), and Polar Knowledge Canada and the Climate Change Preparedness in the North Program of Crown-Indigenous Relations and Northern Affairs Canada. University partners were supported by Natural Sciences and Engineering Research Council of Canada – PermafrostNET (S.G.: NETGP 523228-18), Canada Research Chairs (J.L.B.: 2021-00034; S.E.T.: 2023-00288), and Discovery Grants (J.L.B., D.F., T.C.L., and S.E.T.), and Global Water Futures (Canada First Research Excellence Fund). Northwest Territories Geological Survey contribution 0172. Access to Inuvialuit, Gwich'in, K'asho Got'ine, and Yellowknives Dene lands is gratefully acknowledged. Constructive comments from Yifeng Wang, Michel Allard, David K. Swanson, and an anonymous reviewer improved the manuscript.

## Author contributions
S.V.K. worked with S.A.W., D.F., T.C.L., J.L.B., S.G., S.E.T., P.D.M., H.B.O., J.V.S., N.W., N.J.S., and A.A. to develop the concepts presented in this paper. Permafrost landform data were generated through the Northwest Territories Thermokarst Mapping Collective doi.org/10.1139/as-2023-0009. N.W. and A.S. performed the statistical analyses with guidance from J.L.B., T.C.L., S.G., S.E.T., and S.V.K. N.W., with support from N.J.S., J.V.S., and S.V.K., designed and produced figures with input from all authors. SVK wrote the manuscript with guidance and editorial input from all authors.

## Competing interests
The authors declare no competing interests.
