## [Transparent Peer Review file · Nature Communications]

Permafrost landsystems define regional variability in climate change effects on northern environments

Corresponding Author: Dr Steve Kokelj

Version 0:

Reviewer comments:

Reviewer #1

(Remarks to the Author)

The authors introduce in their manuscript the concept of “permafrost landsystems” which are spatial assemblages of Quaternary landforms (glacial, non-glacial, glacio-lacustrine plains, wetlands...etc.) bearing typical permafrost features (e.g. tundra polygons, thermokarst ponds, retrogressive thaw slumps.. etc. (a list of 28 of them in supplementary table 1, illustrated with nice photographs). Perfectly true in my opinion: landforms and the ecosystems they support provide an excellent mean to map permafrost properties (e.g. types and amounts of ground ice, sensitivities to thaw, susceptibilities for landslides and mass movements). Currently, almost all of the broadscale (e.g. circumpolar) efforts in mapping permafrost conditions using climate data, climate model outputs and remotely sensed surface parameters fail to produce outputs that can inform well on high resolution characterization of permafrost and thaw sensitivity. Those models do not either provide truly useful knowledge for decision and policy makers at the level of a country (like Canada) or a region, even a large one such as NWT. This manuscript therefore proposes an improvement in regional and national approaches of permafrost knowledge acquisition and sharing. Thanks to the work of some of the co-authors (O'Neill et al. 2019), Canada already has a predictive map of ground ice that is unique. It is currently being refined. This paper goes further in the same direction.

I think the manuscript needs major revisions, particularly in tightening the reasoning and focusing the arguments for the use of integrated surveys (“multivariable” here is not the correct word) and ecological mapping. From the start, the reader needs to have a clearer view of how the existing land classification scheme in NWT is structured and what is its terminology. The reader has difficulty to find his way among the ecozones and the ecoregions of various levels of integration. What levels are defined by biomes and climate. What levels are defined on vast areas by physiography. How do they nest into each others. Integrated surveys and ecological land classification started in the 1950s by the CSIRO in Australia. They were adopted in Canada in the 1960s (see Lacate, 1969, pdf joined). The nested concepts of land region, land district, land system, and land types are straightforward and provide a basic framework. A comparable framework is absent from the manuscript and this creates confusion (see my comment at line 280 of the annotated pdf manuscript). Having done contract work in ecological mapping in the late 1970s, my experience tells me that the “permafrost land systems” are the same thing as conventional landsystems, but applied to permafrost landform assemblages.

The paper lacks a clear statement of what are its objectives. As far in the text as line 114, I could not find such a statement. The aims of the paper should appear clearly.

At line 120, the term landform type is introduced. I would say permafrost landform (there are 28 of them that reoccur in the landscape) and save the word “type” for “land type”, which refers to a high level of integration of soil, morphology, vegetation and wetness (but at a larger mapping scale) in land classification procedures. For instance, the nice block-diagrams of figures 1 and 2 which represent assemblages of landforms can be subdivided in land types that have almost uniform combinations of soil, morphology, slopes and vegetation cover. Similarly, block-diagram 3 on figure 3 could be subdivided into three permafrost land types (ridge with ice wedges, wetland with low center polygons, high center polygons). Both in theory and practically, this is the level at which permafrost conditions are nearly uniform. This is the level of integration that would be applicable for designing an infrastructure, for instance.

Some questions that rose in my mind while reading are: are the permafrost land systems mapped over the study region? Is there a map or a data file of them? Or are the land systems a concept that the authors recommend using on an ad hoc basis for specific infrastructure, community or impact assessment projects in the region? Could a gridded climate model be overlaid and integrated with a map of permafrost land systems? These are matters at least for discussion.

The process, or method, applied to run the RDA analysis needs to be explained more simply. See my comments in bubbles and highlighted text on the pdf. I was a bit confused about what variables were used in the RDA. I understood these are only the climate variables and the morphons (i.e. relative elevation values). If so, it is not surprising in my mind that the total explanation of variance is only 23%. Of course, climate regulates the thermal regime and the rates of the processes acting in permafrost. But the surficial geological units (an identification value) that are the main controls on ground ice contents is not a numerical value usable in statistics.

From lines 280 to 364, the five insights can be commented upon in much shorter text. Some basic ideas can be stated more simply. For example, there is less diversity in the discontinuous permafrost zone simply because only palsas and peat plateaus are left as permafrost in the landscape.

Similarly, the discussion is too long and should be more focused.

There is no concluding statement. There is so much value to be added in permafrost mapping in adopting geomorphology and Quaternary through the permafrost landsystem approach.

In general the text could be better integrated, concise, somewhat compacted and less "overtheoretical".

The figures are excellent. I made many comments on the pdf.

Reviewer #2

(Remarks to the Author)

Review of "Permafrost landsystems provide the key to understanding climate change effects on northern terrain and ecosystems"

by Steven V. Kokelj et al.

This study analyzes an extensive dataset on the occurrence of permafrost landforms gathered by the NWT Thermokarst Mapping Collective, in order to bring the wealth of information it contains to a wide audience. The authors' interesting analytical innovation is to apply statistical techniques developed in plant community ecology to a geomorphological dataset, by treating systematic 7.5 x 7.5 km remotely-sensed grid squares as the "plots" and the permafrost landforms in each plot like plant species in community ecology. These statistical techniques are then used to draw inferences about the significance and distribution of permafrost landforms in the study area.

I believe the application of the community-ecology methods to this dataset is both innovative and promising. RDA analysis (the main statistical tool used here) is a powerful technique, and it yielded reasonable results, but nonetheless I have some thoughts about its application.

- 1) My understanding is that RDA is appropriate for response variables with more-or-less normal distributions, while in your data set the response variable was binary presence-absence (as stated in the methods, p. 22 line 21, though the barplots in Fig. 3 imply a semiquantitative scale "trace", ">trace").
- 2) RDA is designed to ordinate the response variables (samples of permafrost landform assemblages) in a space created by the explanatory (environmental) variables. It did this successfully here in the sense that RDA discovered the familiar north-south climatic gradient in permafrost landforms and also a sloping vs. flat landform gradient. It is comforting (but not highly informative) to see these well-known trends confirmed using "big data", and also not surprising that most of the variation in landforms (77%) was actually not explained by available GIS variables. The authors undoubtedly suspected that this was going to be the case beforehand, from their very extensive experience with mapping permafrost landforms; that is probably why they invested so much effort in manual mapping! So my reaction to this would be to recognize RDA's limited usefulness in this context: the explanatory (GIS) variables are statistically significant but have weak explanatory power; RDA actually limits your understanding of permafrost landform assemblages to the 23% of the variation in permafrost landforms that is associated with the environmental (GIS) variables. Unfortunately, the nature of the remaining 77% of the variation unknowable by this method. My response then would be to spend less time on RDA and look at other community ecology methods that might teach us new things. The use of species accumulation curves by level IV ecoregions was a good first step.

Specifically, I was left wanting know see the "Permafrost landform assemblages" defined and described quantitatively. The illustrations in the paper only hint at which landforms occur together in assemblages. Multivariate plant community analysis techniques like cluster analysis, PERMANOVA, and NMDS are designed specifically to discern and describe communities/assemblages. The NWT Thermokarst Mapping Collective dataset is globally unique in its suitability for such analysis. The 7.5 x 7.5 km grid cells are a huge unbiased sample of the permafrost landforms assemblages (and ecoregions are excellent strata to summarize them). I would like the authors to consider using other community ecology methods to shed some light on questions like: What are the natural groups of permafrost landforms? How distinct are the groups and how do they overlap? Are they distinct enough to be mappable? Which "species" (landforms) are "generalists" (occur everywhere and thus are not highly diagnostic of permafrost conditions), and which landforms are very specific and diagnostic of a unique assemblage or diagnostic of unique permafrost conditions? I understand that classification of permafrost landform assemblages, like any classification exercise, could be a time-consuming rabbit-hole to disappear down, but it might also be very fruitful. And you appeared to suggest doing it on p.18 line 16. If you were able to develop a classification of permafrost

landform assemblages, such that the dominant assemblage in each grid cell could be classified, then you would instantly have a 7.5 km pixel resolution map of permafrost landform assemblages for your whole study area. It would undoubtedly be related closely to the ecoregions, but the finer patterns within and between ecoregions could be fascinating and useful. If the authors attempted this and decided it wasn't worth pursuing, please explain why.

An additional benefit to defining concrete landform assemblages would be to give the reader a better picture of what they are. Along similar lines, the "permafrost landsystem" concept can be difficult to grasp. Would a mapped area dominated by one or few permafrost landform assemblages qualify as a "permafrost landsystem"? If so then defining landform assemblages and mapping them by labeling the 7.5 km grid squares would be a great way to make permafrost landsystems more concrete and understandable.

A few other miscellaneous comments:

This study relies heavily on the Level IV Ecoregions by the Ecosystem Classification Group (2008-2013). However, only the 2008 Taiga Shield ecoregion inventory is cited in this article, though ecoregions from several other reports were used.

Most text on the graphics is 6 pt., with some as small as 4 pt. The resolution is good so I was able to read them, but only after enlarging the text on-screen to 200%.

The Methods section begins by saying that permafrost landforms were systematically inventoried for the entirety of the Northwest Territories. But the soon after on p. 22 line 16 it says that systematic observations of these landforms were made from 14 level-IV ecoregions. So it appears that this report is specifically about the subset of 14 ecoregions. Please clarify the connection of this study to the territory-wide inventory.

It would be helpful to provide some more information about the sampling methods, beyond the citation of Kokelj et al. (2023). I understand that you selected 2030 7.5 x 7.5 km squares, which cover 114187 km², and presumably these were restricted to be within the 14 ecoregions. This is just under 10% of the full area of the NWT, and judging from the size the ecoregions on Fig. 2 it appears that the 7.5 x 7.5 km squares covered the entirety of the selected ecoregions. Do you have any more explanation for why you chose these 14 ecoregions? Also, citing Kokelj et al. (2023) and also the barplot in Fig. 3 might imply that you inventoried some landforms by cover classes, but p. 22 line 21 implies only presence/absence of landforms within each grid cell was recorded. I get why you would have done this (presence/absence is much faster), but please clarify. Also please note briefly what imagery was used in the inventory.

p.23 line 662. "Topographic indices" were computed for each 7.5 x 7.5 km grid cell. These appear to me to be indices that are typically computed for pixels (90 or 250 m), and it seems to me that they could vary widely across a 7.5 x 7.5 km grid cell. Please explain how they were generalized for 7.5 x 7.5 km grid cells.

p.23 line 672. "Following standard procedures, we scaled environmental variables for the 7.5 x 7.5 km grid cell framework." It's not clear to me what the standard procedures might be. Z-score across the whole study area?

Supplementary Table 1

Very useful and well done. One comment:

Patterned_Forest_Lowland - the caption seems to indicate that this class was used on both polygonal and string-flark patterns. I hope not, as they are usually distinguishable and, as the caption notes, have different permafrost interpretations. And then later you indicate that you have a separate string fen class.

Supplementary Appendix 1

Well constructed and very helpful. A few comments:

Anderson Plain - "Black arrows" noted in the caption are not visible.

East Bank Hills, Richardson Plateau, Travailant Upland - Retrogressive thaw slumps can difficult to find, especially for the inexperienced eye at the publication scale. Consider arrows to indicate, and possibly mention the resulting turbid lakes.

Reviewer #3

(Remarks to the Author)

This is a well-written paper with sound scientific information. I envision the methodologies established here as being useful in Arctic regions for work that will help educate decision-makers on potential climate-driven landscape changes. I can imagine future applications of this study helping to prioritize areas for future investigations by the Alaska Division of Geological and Geophysical Surveys.

What are the noteworthy results?

- This work provides a significant way to compare permafrost between diverse regions by looking at similarities in landform packages. I think it's a significant broad-scale step in understanding landscape evolution and its relationship to climate change. As noted, it does have its challenges, and field data will improve the process.

Will the work be of significance to the field and related fields? How does it compare to the established literature? If the work is not original, please provide relevant references.

- Yes. It will help us better understand the evolution of permafrost over time on regional scales. As we make land

management decisions, it's essential to have a good understanding of potential landscape changes, which can inform land management decisions and planning for mitigating possible impacts.

Does the work support the conditions and claims, or is additional evidence needed?

- As a first pass, yes. As the authors state, some conditions can't be accounted for with this analysis, and so the methodology could undoubtedly be improved upon, especially with more landforms for the input.

Are there any flaws in the data analysis, interpretation and conclusions? Do these prohibit publication or require revision?

- It would be good to apply this methodology in other areas. The results are dependent on ecozone mapping. Consistency in scale and detail of that mapping when working between areas will impact results for comparison. This work is undoubtedly a significant first step, but how does this work account for those potential inconsistencies?
- What was the reasoning for choosing the grid size? How do the results differ if you select a different grid size?

Is the methodology sound? Does the work meet the expected standards in your field?

- The scientific methodology is sound

Other comments

- Figure 4 is confusing. There is a line labelled "rainfall", but it's unclear what you are trying to convey with that line. The same is true for "percent water," "air temperature," and so on. These are explained in the text, but it's not clear in Figure 4a.
- Figure 4b, there are numerous labels "rock glacier," "patterned fen," etc. It would be helpful if there were lead lines. In its current state, these look like a bunch of words on an image, and it's unclear what you are trying to convey.

Version 1:

Reviewer comments:

Reviewer #1

(Remarks to the Author)

Comments on the revised manuscript "Permafrost landsystems approaches provide the key to understanding regional variability in climate change effects on northern environments"

First, I notice that the title was changed. The original title was "Permafrost landsystems provide the key to understanding climate change effects on terrain and ecosystems". The paper has now a heavier focus on regional variability of both the permafrost landform patterns and, as a consequence, of the climate change impacts on permafrost thawing. As mentioned in the text "the emphasis is that systematic and integrated permafrost mapping provides a critical vehicle for generating new understanding of regional variability of permafrost characteristics". I interpret that such understanding of regional variability is currently inadequately represented by general models (mostly circumpolar in scale) based on spatial climate models and other datasets of terrain variables. I concur with the authors that permafrost landforms being good indicators of permafrost properties, and also the heritage of past climate and ecological changes, they must necessarily be a strong component of the capability for predicting permafrost conditions across landscapes and for predicting the reaction of permafrost to climate change. This is what makes this paper important for the research community and the public.

The authors now state clear objectives at the beginning of the paper. They have added a table (No 2) in the supplementary information showing the existing land classification framework of the NWT, which helps very much to understand the context of application of their proposed landsystem approach. I understand that they consider the level IV ecoregions of this framework as corresponding to the permafrost land systems (this is also stated in their rebuttal letter), although I sense that the text does not seem to be so unequivocal on the matter at some places.

The statistical analysis (RDA) is well explained and the addition of figure 6 showing the loadings of the landforms and of their assemblages across the level IV ecoregions improves very much the communication of the key messages of the paper.

My second reading of the manuscript raised a question in my mind: I think the authors must explain by what scientific process they identified the 28 landform types that are the basic mosaic pieces of landform assemblages that reoccur in the permafrost landscape. They have acquired a unique knowledge of the variety of permafrost landforms in NWT and were finally able to classify them into this set of 28. One or two sentences reporting on their experience would be appreciated. This is important because the variability of occurrence of permafrost landforms in the region reflects the variety of permafrost conditions. These key landforms are also the basic elements of their RDA analysis.

I am adding an annotated manuscript with a few commentaries (mostly suggestions for small changes) for the authors to consider.

Michel Allard

Reviewer #2

(Remarks to the Author)

Comments on "Permafrost landsystem approaches provide the key to understanding regional variability in climate change

effects on northern environments", Steven V. Kokelj et al. Nov. 2025 revision.

by Dr. David K. Swanson (recently retired from U.S. National Park Service, Fairbanks, Alaska)

I thank the authors for their conscientious responses to my and the other reviewers' comments. My few comments on this revised draft of this paper relate mostly to the newly added cluster analysis. I found this analysis to be very interesting and believe it adds a lot to the paper. This analysis provides a statistically-based picture of the composition of the Landform Assemblages. Below are a few specific comments.

Page 5. The introductory section provides a brief overview of the methods that mentions only the Redundancy Analysis. I recognize that the methods overview here must be very short, but given the success of the cluster-heatmap analysis, I think it deserves mention too.

Also on p. 5, this would be a good place to note that the presence/absence data for the 28 landforms by grid cell is your sample of the Permafrost Landform Assemblages. In other words, your list of the landforms present in a cell is the Permafrost Landform Assemblage for that location (given the limitations that the sampling method has, as all of them do). This would help the reader understand how the RDA sheds light on the properties of the Assemblages, because the input data to the RDA is a direct sample of the Assemblages.

Page 12, lines 4-5. The sentence here should read "Figure 6 is a clustered heatmap that explores the natural groupings of permafrost landforms INTO assemblages..." Figure 6 is the place where we get to see which landforms actually group together into assemblages. This info is also available in part from Figure 4, but here the reader must infer the assemblages based on proximity between labels on the diagram, and also their distribution is based only on how they relate to environmental variables (...with only 16.6% + 3.1% = 19.7% of the variability portrayed).

Page 13, 3rd line from the bottom "more gently rising richness curve for the Richardson Plateau". At the scale of Figure 5 at least, the curves for RP and EBH seem to nearly overlap.

Figure 6 is great. I would experiment with the color scheme to show better contrast for the landforms that occur at low frequency.

Page 26, Methods - Statistical approach. The RDA methods are described in great detail, but documentation for the cluster analysis is very brief. The R package "pheatmap" is cited as the method used for the cluster analysis, but this package can be used to plot heat maps made by a variety of clustering methods. Based on the results (Figure 6) it appears that you collapsed the data into a matrix of Level IV Ecoregions (rows) and landforms (columns), with matrix values being percents of the sample cells for each Ecoregion that contained that landform. I could be wrong about this, so please describe the initial matrix that was subjected to clustering, how the distances were calculated, and what cluster method was used. The defaults in pheatmap are Euclidean distance and complete linkage clustering, which is a reasonable choice but not necessarily optimal. You have 3 major clusters, and one of the clusters is quite large with chaining issues. I'd give Ward's method a try and see if it gives better results, if not here then in a future paper. Also, a more fundamental distance measure for the landform clustering might be simply based on the count of 7.5x7.5 km grids where each pair of landforms was both observed. I look forward to reading about this in future publications!

Also in the Methods, you should probably also say which method you used with r function "specaccum". Presumably it was the default ("exact"), but it's best to specify this.

Reviewer #3

(Remarks to the Author)

I believe the authors did a good job of addressing reviewer comments. The clarifications made greatly improve the manuscript and I believe this provides valuable information to assist users at better understanding permafrost conditions at a regional scale. I have not additional comments prior to publication.

Version 2:

Reviewer comments:

Reviewer #1

(Remarks to the Author)

I feel a bit sorry for having suggested so many changes to the authors and maybe for "over debating" in this review. The authors have generously accepted to consider the suggested improvements. The paper is an important contribution to permafrost and environmental science. Particularly, in the current research context on permafrost science, impacts of climate change and useful assessments, this paper demonstrates the fundamental importance of landforms (geomorphology) as key indicators of permafrost properties and dynamic evolution. Similarly inspired work should be expanded across the Subarctic and Arctic regions.

Congratulations!

Michel Allard

Reviewer #2

(Remarks to the Author)

Comments on " Permafrost landsystems define regional variability in climate change effects on northern environments", Steven V. Kokelj et al. 2nd revision (Jan 2026).

I thank the authors for their conscientious responses to my and the other reviewer's comments. The second revision properly addresses the comments that I made on the first revision, and I have no further comments. This paper is an important contribution permafrost science and geomorphology.

David Swanson

Reviewer Reply Summary

We have provided detailed replies to all reviewer comments, most of which have helped to clarify the Manuscript, resulting in editorial modifications and the addition of one analytical figure, and one supplementary Table. Our key objectives and findings, as well as primary analytical techniques and interpretations, remain largely unchanged. We have indicated that three anonymous reviews have improved the clarity and quality of the Manuscript.

We appreciate the detailed comments from the reviewers and their recognition of the value of the conceptual approach and the significant potential of integrated mapping and ecological analysis techniques to advance understanding of variation in permafrost conditions. We address the main comments regarding greater clarity on land classification frameworks and research objectives, and the implementation of additional multivariate methods in the following ways:

1. R1 Ecological land classification schema that we adopt to visualize our data and analyses is clarified. We provide updated references to traditional landsystem approaches, and clearly explain how our study advances these concepts by quantitatively exploring variation in permafrost landform characteristics and developing a coupled conceptual framework to support inferences on intrinsic properties and thaw trajectories. We improve clarity and consistency of language regarding land classifications, and that level IV ecoregions are "a priori" landsystems. We clarify that our landform inventory data is compiled at the scale of a landform assemblage and synthesized at the regional or landsystem scale. We have added a supplementary table to help readers understand how the Northwest Territories (NWT) Land Classification system relates to National frameworks.

2. R1 We explicitly state the aim and objectives of the paper and have added a distinct conclusion section.

3. R1&R2 We better describe our rationale for use of the RDA to explore the landform data and how environmental variables influence their distribution and organization. The technique is valuable in demonstrating that broadscale trends are partially explained by predictor variables, thereby confirming expectations. However, the low explained variation and the graphical patterns illustrate the high degree of diversity in permafrost characteristics across these "known" environmental gradients, which cannot be discovered through heuristic approaches that rely on common predictor variables. This provides a key argument for the value of this novel dataset. The RDA scores and landform richness plots enable quantitative comparison of conditions between level IV ecoregions (a priori landsystems), so both analyses are retained.

In response to R2's comments proposing several analytical methods to further explore the data, we chose to develop a clustered heatmap. This heatmap provides a strong complement and addition to the RDA, exploring common landform assemblages through hierarchical clustering and their influence on distinguishing similarities and differences among a priori landsystems. Other suggestions, including the development of landsystem maps, are beyond the scope of this paper, although we acknowledge that they offer excellent opportunities for future research.

Numerous additional comments are addressed through minor text modifications.

REVIEWER COMMENTS

Reviewer #1 (Remarks to the Author):

The authors introduce in their Manuscript the concept of "permafrost landsystems" which are spatial assemblages of Quaternary landforms (glacial, non-glacial, glacio-lacustrine plains, wetlands,..etc.) bearing typical permafrost features (e.g. tundra polygons, thermokarst ponds, retrogressive thaw slumps.. etc. (a list of 28 of them in supplementary table 1, illustrated with nice photographs). Perfectly true in my opinion: landforms and the ecosystems they support provide an excellent mean to map permafrost properties (e.g. types and amounts of ground ice, sensitivities to thaw, susceptibilities for landslides and mass movements). Currently, almost all of the broadscale (e.g. circumpolar) efforts in mapping permafrost conditions using climate data, climate model outputs and remotely sensed surface parameters fail to produce outputs that can inform well on high resolution characterization of permafrost and thaw sensitivity. Those models do not either provide truly useful knowledge for decision and policy makers at the level of a country (like Canada) or a region, even a large one such as NWT. This Manuscript therefore proposes an improvement in regional and national approaches of permafrost knowledge acquisition and sharing. Thanks to the work of some of the co-authors (O'Neill et al. 2019), Canada already has a predictive map of ground ice that is unique. It is currently being refined. This paper goes further in the same direction.

I think the Manuscript needs major revisions, particularly in tightening the reasoning and focusing the arguments for the use of integrated surveys ("multivariable" here is not the correct word) and ecological mapping. From the start, the reader needs to have a clearer view of how the existing land classification scheme in NWT is structured and what is its terminology.

Response: We appreciate the comment.

Land classification schemes: We clarify the NWT land classification scheme and provide more consistent language, along with a supplementary table that provides the necessary details. We clarify that the paper uses the scheme to stratify and visualize our integrated landform mapping, demonstrating that our permafrost landform inventory approach provides a quantitative basis for exploring regional-scale variability in permafrost conditions. The supporting conceptual framework enables the growing research community to interpret landform data and make inferences about subsurface conditions and the trajectories of thaw-driven change. We have refined our terminology and provided additional background on the NWT land classification scheme and its linkage with national frameworks. This is accomplished through minor modifications in the Introduction, text, and minor labeling adjustments in the figures, as well as some additional information in the methods, including a key supplementary table (2) and references to Lacate (1969).

Clarifying objectives: We have modified the text to clarify the utility and application of permafrost landsystem and mapping approaches. We explicitly state four objectives and the methods applied to achieve them. We have adjusted the text to clarify how the existing land classification scheme in the NWT was utilized to support visualization of the multivariate landform mapping analyses. We emphasized that our goal is to demonstrate that integrated permafrost landform mapping and multivariate analyses methods provide a key opportunity to quantify regional variability in permafrost landscapes, and that permafrost landsystems provides both a mapping and conceptual framework for interpreting results towards better understanding consequences of thaw, upscaling with appropriate

context, and communicating permafrost knowledge across disciplines, knowledge systems, and to the public. We believe this contribution constitutes a substantial advance in quantifying regional variability in permafrost conditions and provides a framework enabling multidisciplinary permafrost science.

The reader has difficulty to find his way among the ecozones and the ecoregions of various levels of integration. What levels are defined by biomes and climate. What levels are defined on vast areas by physiography. How do they nest into each others. Integrated surveys and ecological land classification started in the 1950s by the CSIRO in Australia. They were adopted in Canada in the 1960s (see Lacate, 1969, pdf joined). The nested concepts of land region, land district, land system, and land types are straightforward and provide a basic framework. A comparable framework is absent from the Manuscript and this creates confusion (see my comment at line 280 of the annotated pdf manuscript). Having done contract work in ecological mapping in the late 1970s, my experience tells me that the "permafrost land systems" are the same thing as conventional landsystems, but applied to permafrost landform assemblages.

Response: We appreciate the comment. We clarify that our paper builds on traditional landsystem concepts, and advances a quantitative, permafrost mapping approach to enrich the existing, qualitative land classification schemes adopted in the NWT. We now provide sufficient background on the nested NWT land classification scheme, along with tables and references, to clarify how we utilize it to organize our data and analysis. The analysis demonstrates that integrated mapping enriches our understanding of spatial variation in permafrost conditions within a priori ecological regions. We clarify that an NWT "level IV ecoregion" is conceptually the same as a "landsystem", and that our mapping of landform assemblages and analytical approach allows variation in permafrost conditions within and between these a-priori landsystems to be quantitatively explored. Finally, we demonstrate that permafrost landsystems also offer a conceptual framework to interpret results and communicate more effectively the integrated nature of permafrost, ecosystems, and thaw-driven landscape responses.

The paper lacks a clear statement of what are its objectives. As far in the text as line 114, I could not find such a statement. The aims of the paper should appear clearly.

Response: We have added a clear statement of our goal and objectives.

At line 120, the term landform type is introduced. I would say permafrost landform (there are 28 of them that reoccur in the landscape) and save the word "type" for "land type", which refers to a high level of integration of soil, morphology, vegetation and wetness (but at a larger mapping scale) in land classification procedures. For instance, the nice block-diagrams of figures 1 and 2 which represent assemblages of landforms can be subdivided in land types that have almost uniform combinations of soil, morphology, slopes and vegetation cover. Similarly, block-diagram 3 on figure 3 could be subdivided into three permafrost land types (ridge with ice wedges, wetland with low center polygons, high center polygons). Both in theory and practically, this is the level at which permafrost conditions are nearly uniform. This is the level of integration that would be applicable for designing an infrastructure, for instance.

Response: We removed the word "type". We added an annotation to the Figure showing indicating the landform assemblages.

Some questions that rose in my mind while reading are: are the permafrost land systems mapped over the study region? Is there a map or a data file of them? Or are the land systems a concept that the authors

recommend using on an ad hoc basis for specific infrastructure, community or impact assessment projects in the region? Could a gridded climate model be overlaid and integrated with a map of permafrost land systems? These are matters at least for Discussion.

Response: We appreciate these thoughts and feel that the integrated landform mapping and analyses, considered within a Permafrost Landsystem framework, have exciting potential to build a more quantitative understanding of variability in permafrost conditions and thaw trajectories of northern landscapes.

The raw data in our analyses will be provided with the Manuscript at publication. The gridded climate data is integrated in our analyses as an explanatory variable in the RDA. Several of the points raised by the reviewer are considered in the Discussion.

The process, or method, applied to run the RDA analysis needs to be explained more simply. See my comments in bubbles and highlighted text on the pdf. I was a bit confused about what variables were used in the RDA. I understood these are only the climate variables and the morphons (i.e. relative elevation values). If so, it is not surprising in my mind that the total explanation of variance is only 23%. Of course, climate regulates the thermal regime and the rates of the processes acting in permafrost. But the surficial geological units (an identification value) that are the main controls on ground ice contents is not a numerical value usable in statistics.

Response: We have revised the text and Supplementary table to indicate the explanatory variables used in the RDA clearly. We indicate that surficial geology and modeled ground ice, included as categorical variables in testing, were collinear, derived variables, and did not improve the percentage of variability explained by the model. This finding highlights the importance of the landform inventory in understanding regional-scale variability in permafrost conditions. Broad-scale surficial maps or modeled ground ice are not of sufficient resolution to improve knowledge of variation captured by mapping landforms at the regional scale of inquiry.

One might argue that better or finer-scale surficial maps are a key to improved prediction of permafrost properties (see O'Niell et al., 2024). However, to reiterate, fine-scale surficial products are only available sporadically across vast northern regions. Furthermore, when employed in a heuristic framework, they do not really enable new knowledge to be generated.

From lines 280 to 364, the five insights can be commented upon in much shorter text. Some basic ideas can be stated more simply. For example, there is less diversity in the discontinuous permafrost zone simply because only palsas and peat plateaus are left as permafrost in the landscape.

Response: We appreciate this comment, and numerous others provided as embedded reviewer notes from P12 and 14. However, the "common knowledge" of experts is neither explicit nor accessible in a quantitative manner. The analyses and results presented in this paper provide a basis to improve understanding and quantitatively compare variation in permafrost landscapes at the landsystem scale. The multivariate analyses contain a large amount of information, which we feel is best expressed through explanation tied to place. In relation to R2 comments, we have also produced a clustered heatmap based on all of the mapped grid cells to more deeply support this text. The analysis illustrates novel ways to quantitatively assess and visualize similarities and differences in permafrost conditions

between a-priori level 4 ecoregions (landsystems), and explore how permafrost landforms assemblages combine to determine the differences amongst them.

Similarly, the Discussion is too long and should be more focused.

Response: We have condensed parts of the Discussion while elaborating on a few points and questions highlighted by the reviewer in a previous comment.

There is no concluding statement. There is so much value to be added in permafrost mapping in adopting geomorphology and Quaternary through the permafrost landsystem approach.

Response: Thank you, we have added a concluding paragraph.

In general the text could be better integrated, concise, somewhat compacted and less "overtheoretical".

Response: We have considered this comment in revising the Discussion.

The figures are excellent. I made many comments on the pdf.

Response: We are glad the reviewer appreciated the figures.

Specific comments in annotated PDF. Reviewer 1

P2L36. not clear what is meant by "multivariable" landform inventory. In fact ecological land classification consists in mapping units of territory at different scales of integration of landforms, soil, végétation, climate etc. (ex. region, district, system, type, phase..). They are integrated surveys, an approach to resources mapping developed since the 1950s. In Canada, see Lacate, 1969). As I read in the paper, typical permafrost landforms (a list of 28 of them, as insupplementary file 1) were counted in a number of map pixels (not clear if it is in whole land systems) and the classification tested in a RDA with a limited number of climate variables.

Reply: To clarify the reviewer's initial comment, we have removed the word "multivariable" and adjusted the text to "Here, we analyze an inventory of permafrost landforms from across of northwestern Canada ..."

We have also now clarified that permafrost landforms were assessed in all grid cells (3,290) from the 14 level 4 ecoregions, and a subset of these (2,030) was input into the RDA. Primary climate and terrain/environmental variables were used as explanatory variables in the final RDA.

Rationale and strengths of the approach are described in our general comments, above.

P2L37. assemblages by definition are not homogenous, therefore the mapped permafrost properties will also be spatial assemblages. Differences in "superficial" permafrost landforms and permafrost temperature regime will be noticeable at the scale of ecoregions because they have different climate conditions by definition.

Reply: We agree with the reviewer's comment and have expressed the integrated nature of permafrost environments in our text, which is the essence of permafrost landsystems as a conceptual framework. The approach in the paper enables spatial landform assemblages to be evaluated quantitatively. We

explore variation in the landform data stratified by level IV Ecoregions, conceptually similar to landsystems as expressed in the Ecosystem Classification Group (2007). The hierarchy in the NWT Classification system and its relation to the national framework are referenced directly to a source figure in Ecosystem Classification Group (2007) and summarized in Supplementary Table 2.

P2L39. Quaternary and Holocene history.

Reply: We keep our text more accessible to a broad community and have expressed these two ideas more generally as "geological and climate legacies."

P2L41. maybe make a clearer distinction between variability within land systems (defined by internal geomorphic composition) and variability between land systems of different ecoregions and ecozones which have different climate conditions along the gradient from Subarctic to Arctic). There are local trajectories within systems dependant mostly on soils for impacts and climate-driven regional trajectories to be understood for broader adaptation.

Reply: We appreciate the comment and have made a minor adjustment on Figure 3 to clarify this point.

P2L49. maybe be more specific in the paper. Engineering and infrastructure issues are often at the local scale. Land use planning and policy making are regional in scope. Ways of knowing refer to traditional knowledge of the land by Northeners.

Reply: We appreciate the comment and consider this in our revisions throughout. A minor editorial modification establishes the link between landsystem concepts and decision-making in the abstract; however, word limits constrain further elaboration.

P3. Good general Introduction

Reply: We appreciate the reviewer's comment.

P3L71. I think the O'Neill et al. 2019 model is different from the other ones. In comparison with them it is based on surficial geology and Quaternary history. It better predicts ground ice conditions, hence thermokarst. Most other ones are based on data from climate model outputs and remotely sensed parameters. What you propose here, to me, is different from climate-model based systems and better suited for support national, regional and local decision-making for adaptation.

Reply: We appreciate that the reviewer's comment on O'Neill et al 2019. All of the products referenced in the list are heuristic in nature. While the logic behind O'Neill et al. (2019) aligns well with landsystem concepts, its manifestation summarizes existing knowledge in the rule-based geomatics framework. We consolidate these references to highlight several broadscale, permafrost-related outputs that, despite their inherent limitations in generating new knowledge, collectively shape our understanding of the permafrost environment and the consequences of thaw. The statement sets the context for regional-scale permafrost mapping and the generation of new knowledge within a quantitative framework.

Regarding the final part of the R1 comment, we adjusted the text to emphasize the utility of empirically-driven permafrost landform mapping, analysis, and interpretation within a permafrost landsystems framework, aiming to advance understanding of variation in permafrost conditions necessary for national, regional, and local decision-making.

P3L72. true in my opinion. Experience and perceptions of local people and public administrations diverge from model-based circumpolar maps of permafrost conditions

Reply: We appreciate the reviewer's view on this contextual statement, which helps make a case for permafrost landsystems as an important analytical and conceptual framework.

P3L84. I agree. Wide scale predictive models are of little practical use because they are not based on adequate knowledge of the terrain and subsurface conditions that can be captured only at the scale of landforms and mappable surficial geological units.

Reply: We appreciate the reviewer's comment.

P3L84 Why renewed?

Reply: We have changed this to "growing" rather than "renewed". The citations here provide examples of mapped variation in permafrost landforms over large areas. In two cases, the papers demonstrate incongruence between modeled outputs and systematically collected permafrost landform data.

P4L93. I suggest here that you revisit the concepts of ecological mapping and land classification that were widely applied in Canada in the 1970s and 1980s. see Lacate, 1969. Although they were not used in permafrost regions, they still hold very well. (Region, district, systems, types, phase)

Reply: We thank the reviewer and have added reference to Lacate 1969. As indicated above, we now provide additional context on traditional land classification systems by adding a concise narrative, a new supplementary table 2, and references, clarifying how the NWT land classification framework relates to the legacy of past approaches (e.g., Lacate, 1969), and fits within the national/continental framework (Table 1 in Ecosystem Classification Group, 2007).

We also add text in the Methods to clarify the NWT land classification system and its linkages to ideas presented by Lacate and others. We also clarify that the approach in this paper "extends Landsystem thinking through an explicitly permafrost lens, applying a conceptual tool to support mapping and advance discourse on critical permafrost issues."

P4L111.integrated mapping = better understanding of the functioning of ecosystems. Can lead to better appraisal of environmental and social consequences of climate warming and thawing permafrost.

Reply: We agree, and this point is expressed in the Introduction and now explicitly in the Fig 3 caption.

P4L111. At this point, it is unclear what the objectives of the paper are. In the following paragraphs, you explain the concept of permafrost land systems and you attempt to demonstrate its "applicability" rather than a real or potential application. The concept of land system in ecological land classification is 60-70 years old. You are ably extending it to permafrost regions.

Reply: We have adjusted text to indicate the overarching aim, specific study objectives, and explicit linkages to analytical methods. We clarify that this paper aims to extend traditional landsystem concepts to permafrost regions, and develop coupled mapping and analyses to characterize and contrast permafrost landscapes quantitatively within a landsystem framework.

To our knowledge, this is the first time that systematic permafrost landform inventories have been analyzed quantitatively to evaluate the composition and variability of the permafrost landscape at regional scales. Furthermore, we highlight the permafrost land system framework as a conceptual tool

for interpreting spatial landform patterns and inferring ground-ice, soil, and temperature conditions, as well as potential trajectories of thaw-driven change. While this may be known to the expert reviewer, we feel it is important to express this in quantitative terms to the growing diversity of scientists, planners, and policymakers interested in understanding variation in permafrost environments.

P5L120. so a 7.5 X 7.5 km grid was laid over the study region and you counted the permafrost landforms in 2030 cells. Were the cells selected at random or by some stratified sampling among ecoregions? 2030 represent what % of the total number grid cells over the study region? how can you affirm that the sample is representative of the whole set?

Reply: We have adjusted the text to clarify that our RDA analyses (mapped cells and derived environmental/predictor variables) comprises a subsample of 7.5x7.5 grid cells (1959) from 14 level 4 ecoregions, which are conceptually similar to "landsystems". Descriptive summaries, landform richness curves, and a new clustered heatmap depicting landform and landsystem associations comprise all grid cells (3,402) within the 14 level 4 ecoregions.

We clarify that our selected ecoregions aim to evaluate conditions encountered across the NWT. We did NOT randomly sample the entire 1.3 million km² area of the NWT. In the Introduction and methods, we clarify that the ecoregions selected represent a sample of climate and geological endmembers found across the continental-scale ecoclimate gradient in the NWT, allowing us to demonstrate the application and utility of our quantitative approach. The NWT Ecoregion hierarchy used to stratify our data is now also shown in Supplementary Table 2.

P5L123. I see in text and supplementary table 2 that only the climate variables and the morphons were used in the RDA (am I correct?). These are numerical variables (ex. temperature values, elevation differences). Therefore the geological variables (ex. sediment type (not numerical) and classes of ground ice contents (not truly numerical like volumetric ice contents) were not applied in the RDA). Am I correct? this is not crystal clear in the text, one has to search for these precisions in the text and tabl

Reply: We indicate that only primary numerical variables or those derived directly from quantitative datasets are used as predictor variables. Following careful consideration among the diverse Authorship team, we chose not to include hueristically-derived broadscale modeling products as predictor variables because they typically reflect an integration of other broadscale datasets, which are often highly correlated. Regardless, we tested categorical data from several derived or generalized datasets as predictor variables in the RDA, including surficial geology and predicted ground-ice and permafrost probability. We found that their inclusion had little to no impact on the % of variability explained in the observational data. We indicate this highlights the importance/utility of systematic landform mapping for understanding variability in permafrost conditions.

P5L124. Ecozone

Reply: Our stratification for broadscale analysis is at the level 2 ecoregion, or ecozone. We adjust the text and figures accordingly. We reference to the NWT classification system, as well as mapping scales and terminological equivalents in Supplementary Table 2.

this is a very complex land classification system. The international reader will have much difficulty to understand it. Where do excatly permafost land systems fit in that scheme. Have you mapped the land

systems at a scale corresponding to diagrams of figure 1 and 2? or are they concepts to be used at variable scales? is the concept primarily intended for extracting local permafrost information from maps of ecoregions?

Reply: We have provided Supplementary Table 2 that summarizes the NWT classification scheme and how the levels fit within national frameworks. We clarify that level 4 ecoregions are conceptually similar to landsystems, explain this in the methods, and provide a reference and table to clarify this. We clarify that our data is generated at the grid cell scale, and is analyzed and summarized at the landsystem scale for regions ranging from 10^3 to 10^4 km². We also demonstrate that analyzing systematic landform inventory data alone has potential to derive permafrost landsystems. Our analysis highlights the potential of systematically collected permafrost landform data and analyses to enrich or even restructure existing schemes on a quantitative basis, thereby enabling the generation of new knowledge about permafrost terrain conditions. We also indicate that the approach can upscale local results to broader regions with quantitative context.

P5L131. mabe be more specific here: by permafrost conditions one means ground ice types and contents, range of temperature profiles, carbon (organic matter) contents ??

Reply: We clarify this through a minor modification of the text.

P5L134. "integrated"

Reply: The suggested editorial modification has been implemented.

P5L136. If I understand well, the aims of this paper is to explain the approach and to propose to use the land system concept for those objectives. But are the mapping units (permafrost land systems) available on maps/file?

Reply: We have adjusted the text to clarify that we advance a permafrost landform inventory/mapping approach to quantify variability in permafrost landscapes at the "landsystem scale", and a conceptual framework (permafrost landsystems) within which to interpret the data. We emphasize that combining a quantitative approach with a conceptual framework can support a more robust appraisal of the environmental and societal consequences of climate change and thawing permafrost at regional scales.

The inventory results of mapped grid cells for the ecoregions will be provided upon publication.

P7L166. I think the ground ice in the glaciolacustrine sediments must be segregated.

Reply: Yes, we agree and have modified the text accordingly.

P8 L 177. as in previous land classification schemes, permafrost land systems are themselves assemblages of higher scale assemblages of "permafrost land types". In fact the true integration, i.e. one soil type - a dominant pattern of permafrost landforms (metre to decameter scale), one class of ground ice content, one class of organic layers thickness - is at the land type level.

Reply: We appreciate the reviewer's views.

Minor modifications to the captions/descriptions of Figures 1-3 clarify that assemblages and land types are synonymous. The reviewer's point is expressed through the graphic.

P9L223. let's not forget that ecoregions and ecozones reflect climate conditions... if the subarctic Taiga plain is a vast ecotone (i.e. transitional), then it poorly represents regional climate. It is rather physiographic.

Reply: The high subarctic Taiga plains are a level 3 ecoregion, comparable to an Ecoprovince (as defined by the Canada Committee on Ecological Land Classification) or an ecoclimate region (Ecoregions Working Group, 1989). Level 3 ecoregions are defined by regional climate differences within Level 2 ecoregions (Ecozones). Terminology has been clarified in the text and linked to Supplementary Table 2 and a referenced summary table in Ecoregions Classification Group, (2007).

Our data and analysis show that hard lines on geology, ecoregion, and land cover maps are often fuzzy boundaries. The data and analyses provided here offer a transparent, accessible, and quantitative approach to explore a priori land classification schemes, the outputs of heuristic mapping products, and exciting opportunities to learn new things about variation in permafrost landscapes and existing classifications that are products of mapping overlays integrated in a GIS framework.

P10L248. i.e. applying only broadscale spatialized climate and modelled data without incorporating knowledge of surficial geology and climate history

Reply: We agree and have tweaked the sentence slightly to be more explicit. However, the salient point here is the importance of advancing empirical approaches to generate new knowledge of variation in permafrost conditions.

P10L251. this rather indicates that geology and ecological conditions need to be included in spatial analyses and mapping through systematic landform inventories ...

Reply: We thank the reviewer for this comment and have made the necessary adjustments.

P12L280. The paper needs more clarity. The recommended land units for understanding permafrost in the paper are land systems. In fact, the true integration of ecological components is at the land type level. And the paper comments the results of the applied methodology at the ecoregion level. That creates confusion

Reply: We thank the reviewer for their perspective. We indicate that the mapping inventory of landforms identifies assemblages and demonstrates that their organization at the landsystem scale can be quantitatively assessed to better understand regional-scale variability. In the abstract, manuscript text, and figures, we argue that the true integration of soil, ground ice, and ecological components is found in landform assemblages or "land types," and quantitatively assessing their patterns, here at the 7.5x7.5 km grid cell provides data for quantitative analysis and regional-scale understanding of permafrost conditions.

The emphasis is that systematic and integrated permafrost landform mapping provides a critical vehicle for generating a new understanding of regional variability in permafrost characteristics. This approach moves beyond heuristic/expert knowledge-based classification frameworks or geomatics overlay activities to portray landscape regions. Our mapping approach provides a first attempt to generate systematic, quantitative regional-scale permafrost knowledge. A linked, geologically based conceptual framework (permafrost landsystems) enables information to be transferred across scales and communicates the impacts of thaw-driven change to broad audiences.

P12L288. Yes. likely true everywhere in the World, not only in permafrost regions. But this is important to recall.

Reply: We appreciate the reviewer's comment and agree.

P13L322. to be expected. Permafrost is a geological and a climatic phenomenon. The two are inseparable in permafrost science.

Reply: We appreciate the reviewer's comment and agree.

P14L341. normal because ecotones are by definition transition zones where systems change and merge from one to another, thus creating diversity.

Reply: We appreciate the reviewer's comment and agree.

P14L351 this is self evident! the only landforms left in the landscape are decaying palsas and peat plateaus. Often the discontinuous permafrost zone has more non-permafrost than permafrost terrain (as low as 1-5% area)

Reply: We appreciate the reviewer's comment. However, although this may be self-evident to a disciplinary specialist, we feel that the point is important for the broad audience that requires knowledge of permafrost and permafrost change.

P14 L360. I suggest that those five paragraphs of insights be reduced in length and be explained in simple sentences (maybe in a less theoretical style?)

Reply: We chose to keep this section to its original length and have added an additional figure (Figure 6, clustered heatmap) to quantitatively visualize "self-evident" knowledge. We agree that an expert may expect the broadscale trends expressed in our analysis. However, these comments prompt recognition that expert knowledge reflects a highly variable, non-explicit, and often proprietary understanding of the permafrost landscape held by a small group within a rapidly growing community of multidisciplinary researchers, planners, and policymakers interested in permafrost and ecosystem change. This expert knowledge is typically summarized through general statements and rules of thumb, supported by schematic representations of the environment. Expert knowledge, summarized through heuristics applied within a geomatics framework, manifests as rigid boundaries on maps, and the quality and resolution of derived data products constrain outputs. Rule-based models reflect decisions based on an expert knowledge base, providing an economical way to portray what we think we already know. The sources of error and uncertainty are typically not explicit, and variability cannot be accurately assessed.

P15L366. Figure 5. Maybe explain on the graph scale label or in the caption what is "Richness". Number of observed (recurring) landforms (from the list of 28 inventoried) counted in the grid cells

Reply: Minor editorial modification is applied.

P16. The Discussion is too long and takes the reader in many different directions. Could be shortened in half and address only the central topic.

Reply: We appreciate the reviewer's perspective. Although we have removed a paragraph discussing landform and ecosystem evolution, based on the reviewer's previous comments, we feel that the

Discussion is a reasonable length and covers key topics we wish to highlight in this paper. We highlight the constraints of existing qualitative and heuristic products that portray regional-scale permafrost conditions, and emphasize the exciting potential of integrated permafrost landform mapping and quantitative analyses to gain new insights into regional-scale variation in permafrost conditions. We highlight permafrost landsystems as a mapping and analytical approach to generate and visualize new knowledge about regional variability in permafrost conditions. We finally describe how the approach serves as a conceptual framework for interpreting data, making inferences, and communicating permafrost knowledge across disciplines, to decision-makers, and across knowledge systems. As described above, our text is targeted toward a broad, non-specialized audience with a vested interest in the permafrost sciences, as appropriate for the audience of Nature Communications.

P16L393. Permafrost landforms and thermokarst landforms manifesting the consequence of thaw are regrouped in land patterns or spatial assemblages that we called permafrost land systems, which are indicative of the fundamental permafrost properties in a region.

Reply: We implemented a minor editorial modification to the first sentence of the Discussion to address this comment.

P16L401. a question for Discussion: would simply use surficial geology map at a proper scale as base layer and superimpose permafrost and thermokarst landforms make a product of equal or better value?

Reply: This integrated approach is certainly worthy of further exploration; however, it falls beyond the scope of this paper. A major challenge is to obtain surface geology maps at appropriate and consistent scales to address questions of variability in permafrost conditions across a territorial scale (10^6 km^2) and to ensure consistent comparison of ecoregions (landsystems), gradients, and patterns across regional scales. The RDA shows that if the best surficial map across all landsystems of interest is used, it is at an insufficient scale to explain much of the variability in the mapping data. So for characterizing regions within a comparable, quantitative framework, the mapping data reveals much new information.

P16L414. is RDA the proper predictive method? not sure because surficial geology units are not quantitative variables?

Reply: The RDA is a flexible tool for examining which environmental gradients explain the variability in how permafrost features organize and where they occur. Adding surficial geology as a categorical variable had little impact on the explanatory power of the RDA.

P16L418. could the problem of heterogeneity be solved by integrated mapping at a larger scale (scale of permafrost land types). I understand however that NWT is vast. But that level of integration could be used for specific studies (communities, infrastructure projects, environmental assessments..)

Reply: We agree that heterogeneity at the regional (landsystem scale) can be described quantitatively through systematic/integrated mapping. We make a minor modification to emphasize this gap in the Discussion, its application for comparing conditions between regions, and its use in community and infrastructure planning projects for scoping Environmental Assessments.

P17L432. you need to say how the distribution of carbon and its contents are related to the permafrost land systems by association with specific landforms and through ecological history.

Reply: This point was addressed in the following paragraph, which the reviewer has suggested we delete.

P17L443. This whole paragraph is not necessary in the paper

Reply: The linkages made in this paragraph are implicit to the expert reviewer but they are not completely understood by many scientists studying permafrost systems. We have deleted most of the paragraph but retained the key point that field-based investigations and monitoring of landform assemblages are critical for guiding inferences from the spatial patterns determined by systematic landform mapping.

P18L474. Models based on climate projections, depicting permafrost distribution and predicting upcoming degradation would benefit to be applied onto permafrost land system maps to truly represent permafrost conditions and be of practical use.

Reply: We thank the reviewer and adjust the text slightly to make this point.

P19L482. this refers to people living in the North and to practitioners (e.g. engineers, land use planners). But the paper does not allude to the variable perceptions and experiences of people across the vast region of study.

Reply: Variable perceptions of people's understanding of permafrost is not a focus of this paper. Our point here is simply that there are variable perceptions and experiences regarding people's relationship with and knowledge of permafrost. The conceptual framework offered by permafrost landsystems has significant potential to help foster communication of knowledge across different groups towards a common or better understanding of conditions, trajectories of change, and consequences.

P18L488. Indigenous?

Reply: Yes and we have added an appropriate reference.

Reviewer #2 (Remarks to the Author):

Review of "Permafrost landsystems provide the key to understanding climate change effects on northern terrain and ecosystems"
by Steven V. Kokelj et al.

This study analyzes an extensive dataset on the occurrence of permafrost landforms gathered by the NWT Thermokarst Mapping Collective, in order to bring the wealth of information it contains to a wide audience. The authors' interesting analytical innovation is to apply statistical techniques developed in plant community ecology to a geomorphological dataset, by treating systematic 7.5 x 7.5 km remotely-sensed grid squares as the "plots" and the permafrost landforms in each plot like plant species in community ecology. These statistical techniques are then used to draw inferences about the significance and distribution of permafrost landforms in the study area.

I believe the application of the community-ecology methods to this dataset is both innovative and promising. RDA analysis (the main statistical tool used here) is a powerful technique, and it yielded reasonable results, but nonetheless I have some thoughts about its application.

1) My understanding is that RDA is appropriate for response variables with more-or-less normal

distributions, while in your data set the response variable was binary presence-absence (as stated in the methods, p. 22 line 21, though the barplots in Fig. 3 imply a semiquantitative scale "trace", ">trace").

Reply: We appreciate the reviewer's comment. We agree that the RDA approach is helpful and – as described in our responses above – yielded illuminating results. In response to this comment and the following comments, we have bolstered our statistical approach by adding analyses, as described below. These additional analyses reinforce and add to the RDA outputs. We have additionally adjusted Fig. 3 and plotted only presence/absence on the bar plot to avoid confusion.

2) RDA is designed to ordinate the response variables (samples of permafrost landform assemblages) in a space created by the explanatory (environmental) variables. It did this successfully here in the sense that RDA discovered the familiar north-south climatic gradient in permafrost landforms and also a sloping vs. flat landform gradient. It is comforting (but not highly informative) to see these well-known trends confirmed using "big data", and also not surprising that most of the variation in landforms (77%) was actually not explained by available GIS variables.

Reply: We appreciate the reviewer's assessment and agree that this analysis supports general, known trends. The data visualization shows substantial variation in permafrost site scores, with a significant portion unexplained by the primary environmental variables. In addition to quantifying the influence of environmental variables on landform patterns and combinations, the RDA illustrates significant variability across the landscape that heuristic approaches and spatial modeling cannot capture. In essence, the fact that the RDA uncovered that the climate variables commonly used to predict permafrost distribution and thaw trajectories explain only part of the variation is a benefit of the analysis. We describe this in our responses above, and also now add some text to the end of the second paragraph of the Discussion to better explain the importance of this finding.

The authors undoubtedly suspected that this was going to be the case beforehand, from their very extensive experience with mapping permafrost landforms; that is probably why they invested so much effort in manual mapping! So my reaction to this would be to recognize RDA's limited usefulness in this context: the explanatory (GIS) variables are statistically significant but have weak explanatory power; RDA actually limits your understanding of permafrost landform assemblages to the 23% of the variation in permafrost landforms that is associated with the environmental (GIS) variables. Unfortunately, the nature of the remaining 77% of the variation unknowable by this method. My response then would be to spend less time on RDA and look at other community ecology methods that might teach us new things. The use of species accumulation curves by level IV ecoregions was a good first step.

Reply: Thanks for this comment. Please see our response above, describing how the "limitation" of the RDA is also illuminating, and our description below of additional analyses. We agree that the RDA confirms general broadscale trends that are known, but are not quantified over regional scales. In this regard, the RDA demonstrates the power of the mapping dataset and highlights challenges with conventional approaches to classification and map drawing. Broad-scale spatial modeling products shape our understanding of permafrost conditions and persist as the best input datasets for various broad-scale applications. Stratifying RDA scores by level IV ecoregions (Fig. 5) was an obvious continuation of the RDA, to demonstrate a quantitative method for visualizing differences amongst regional-scale ecoregions (a priori landsystems). We appreciate the reviewer's comment on species (landform) accumulation curves as an additional step in understanding and quantifying regional variability.

Specifically, I was left wanting know see the "Permafrost landform assemblages" defined and described quantitatively. The illustrations in the paper only hint at which landforms occur together in assemblages. Multivariate plant community analysis techniques like cluster analysis, PERMANOVA, and NMDS are designed specifically to discern and describe communities/assemblages. The NWT Thermokarst Mapping Collective dataset is globally unique in it's suitability for such analysis. The 7.5 x 7.5 km grid cells are a huge unbiased sample of the permafrost landforms assemblages (and ecoregions are excellent strata to summarize them). I would like the authors to consider using other community ecology methods to shed some light on questions like: What are the natural groups of permafrost landforms? How distinct are the groups and how do they overlap? Are they distinct enough to be mappable? Which "species" (landforms) are "generalists" (occur everywhere and thus are not highly diagnostic of permafrost conditions), and which landforms are very specific and diagnostic of a unique assemblage or diagnostic of unique permafrost conditions? I understand that classification of permafrost landform assemblages, like any classification exercise, could be a time-consuming rabbit-hole to disappear down, but it might also be very fruitful. And you appeared to suggest doing it on p.18 line 16. If you were able to develop a classification of permafrost landform assemblages, such that the dominant assemblage in each grid cell could be classified, then you would instantly have a 7.5 km pixel resolution map of permafrost landform assemblages for your whole study area. It would undoubtedly be related closely to the ecoregions, but the finer patterns within and between ecoregions could be fascinating and useful. If the authors attempted this and decided it wasn't worth pursuing, please explain why.

An additional benefit to defining concrete landform assemblages would be to give the reader a better picture of what they are. Along similar lines, the "permafrost landsystem" concept can be difficult to grasp. Would a mapped area dominated by one or few permafrost landform assemblages qualify as a "permafrost landsystem"? If so then defining landform assemblages and mapping them by labeling the 7.5 km grid squares would be a great way to make permafrost landsystems more concrete and understandable.

Reply: We agree that there is much opportunity to further explore the novel dataset using the cross-disciplinary approaches that we have employed. We addressed the reviewer's useful input by developing a clustered heatmap (Fig. 6) to explore common assemblages through hierarchical clustering and to visualize their influence on distinguishing similarities and differences among level IV ecoregions (landsystems). Each row in the clustered heatmap essentially reflects a landform fingerprint distinct to each a-priori ecoregion. While we only explored a couple of methods from community ecology analysis, they have demonstrated the utility and opportunities for adopting them in future research.

The potential for further exploration of this dataset, or of the entire NWT once complete, is tremendous. It offers significant opportunities to learn about common assemblages, their patterns of variation across landscapes, and provides an exciting and relevant research avenue for researchers to explore. We agree that mapping the assemblages is a good idea and will be a focus of future research.

A few other miscellaneous comments:

This study relies heavily on the Level IV Ecoregions by the Ecosystem Classification Group (2008-2013). However, only the 2008 Taiga Shield ecoregion inventory is cited in this article, though ecoregions from several other reports were used.

Reply: We have added citations to the remaining Ecoregion reports.

Most text on the graphics is 6 pt., with some as small as 4 pt. The resolution is good so I was able to read them, but only after enlarging the text on-screen to 200%.

Reply: The figures conformed to Nature guidelines. We have increased font size where possible.

The Methods section begins by saying that permafrost landforms were systematically inventoried for the entirety of the Northwest Territories. But the soon after on p. 22 line 16 it says that systematic observations of these landforms were made from 14 level-IV ecoregions. So it appears that this report is specifically about the subset of 14 ecoregions. Please clarify the connection of this study to the territory-wide inventory.

Reply: We have clarified what ecoregions were sampled and the rationale at several places within the Manuscript and in the methods.

It would be helpful to provide some more information about the sampling methods, beyond the citation of Kokelj et al. (2023). I understand that you selected 2030 7.5 x 7.5 km squares, which cover 114187 km², and presumably these were restricted to be within the 14 ecoregions. This is just under 10% of the full area of the NWT, and judging from the size the ecoregions on Fig. 2 it appears that the 7.5 x 7.5 km squares covered the entirety of the selected ecoregions. Do you have any more explanation for why you chose these 14 ecoregions? Also, citing Kokelj et al. (2023) and also the barplot in Fig. 3 might imply that you inventoried some landforms by cover classes, but p. 22 line 21 implies only presence/absence of landforms within each grid cell was recorded. I get why you would have done this (presence/absence is much faster), but please clarify. Also please note briefly what imagery was used in the inventory.

Reply: We clarify the sampling design and the rationale for choosing the 14 level IV ecoregions to sample. We clarify that we use presence absence data in this paper. Presence/absence is available for all landforms, and it enables patterns of variation to be explored in a manner that meets the paper objectives. We have modified the plate on figure 3 to include only presence absence data to avoid confusion. We have elaborated the methods provided in this manuscript and also point authors to Kokelj et al 2023 and references within, including Open reports that summarize in detail the inventory methods. Furthermore we clarify that Sentinel-2 imagery (2016–2017; 10 m resolution) was used for this inventory and now clearly indicated in the methods.

p.23 line 662. "Topographic indices" were computed for each 7.5 x 7.5 km grid cell. These appear to me to be indices that are typically computed for pixels (90 or 250 m), and it seems to me that they could vary widely across a 7.5 x 7.5 km grid cell. Please explain how they were generalized for 7.5 x 7.5 km grid cells.

Reply: We clarify that we used the variation of "geomorphons" within a gridcell rather than the 'mean'. The intention was to have a metric that separates grid cells with a larger number of different topography types from more homogeneous ones, providing an intuitive index for topographic complexity. Details are in Supplementary Table 3.

p.23 line 672. "Following standard procedures, we scaled environmental variables for the 7.5 x 7.5 km grid cell framework." It's not clear to me what the standard procedures might be. Z-score across the whole

study area?

Reply: Scaling was applied in the RDA function for environmental variables. The sentence was adjusted to improve clarity.

Supplementary Table 1

Very useful and well done. One comment:

Patterned_Forest_Lowland - the caption seems to indicate that this class was used on both polygonal and string-flark patterns. I hope not, as they are usually distinguishable and, as the caption notes, have different permafrost interpretations. And then later you indicate that you have a separate string fen class.

Reply: We removed the confusing extra narrative for Pattern_Forest_Lowland, which aims to capture forest patterns and their relation to polygonal terrain, sometimes visible in low subarctic alluvial deposits.

Supplementary Appendix 1

Well constructed and very helpful. A few comments:

Anderson Plain - "Black arrows" noted in the caption are not visible.

East Bank Hills, Richardson Plateau, Travaillant Upland - Retrogressive thaw slumps can difficult to find, especially for the inexperienced eye at the publication scale. Consider arrows to indicate, and possibly mention the resulting turbid lakes.

Reply: We have added arrows to support readers understanding of imagery patterns.

Reviewer #3 (Remarks to the Author):

This is a well-written paper with sound scientific information. I envision the methodologies established here as being useful in Arctic regions for work that will help educate decision-makers on potential climate-driven landscape changes. I can imagine future applications of this study helping to prioritize areas for future investigations by the Alaska Division of Geological and Geophysical Surveys.

Reply: Thank you. We appreciate the reviewer's comment.

What are the noteworthy results?

- This work provides a significant way to compare permafrost between diverse regions by looking at similarities in landform packages. I think it's a significant broadscale step in understanding landscape evolution and its relationship to climate change. As noted, it does have its challenges, and field data will improve the process.

Reply: We appreciate the reviewer's comment.

Will the work be of significance to the field and related fields? How does it compare to the established literature? If the work is not original, please provide relevant references.

- Yes. It will help us better understand the evolution of permafrost over time on regional scales. As we make land management decisions, it's essential to have a good understanding of potential landscape

changes, which can inform land management decisions and planning for mitigating possible impacts.

Reply: We appreciate the reviewer's comment.

Does the work support the conditions and claims, or is additional evidence needed?

- As a first pass, yes. As the authors state, some conditions can't be accounted for with this analysis, and so the methodology could undoubtedly be improved upon, especially with more landforms for the input.

Reply: We appreciate the reviewer's comment.

Are there any flaws in the data analysis, interpretation and conclusions? Do these prohibit publication or require revision?

- It would be good to apply this methodology in other areas. The results are dependent on ecozone mapping. Consistency in scale and detail of that mapping when working between areas will impact results for comparison. This work is undoubtedly a significant first step, but how does this work account for those potential inconsistencies?

Reply: We appreciate the reviewer's comment. Base imagery and scale will affect the comparability of the data with studies from other regions. However, we are unaware of regional-scale datasets that are comparable because studies typically assess one or two feature types. We are hopeful that this work will spur others to map permafrost features more broadly, which would enable the expansion of the approach. There is the possibility to extend the mapping across Canada, or at least to areas of concern.

- What was the reasoning for choosing the grid size? How do the results differ if you select a different grid size?

Reply: The grid size was chosen because they represent quartiles of a previous effort to map mass wasting using 15x15 km grid cells across northwestern Canada (Kokelj et al 2017), and they were deemed granular enough to provide fine-scale patterns, within a mapping framework that could be completed by trained mappers over a reasonable timeframe

(Kokelj, S. V., Lantz, T. C., Tunnicliffe, J., Segal, R. & Lacelle, D. Climate-driven thaw of permafrost preserved glacial landscapes, northwestern Canada. *Geology* 45, 371–374 (2017).

Is the methodology sound? Does the work meet the expected standards in your field?

- The scientific methodology is sound

Reply: We appreciate the reviewer's comment.

Other comments

- Figure 4 is confusing. There is a line labelled "rainfall", but it's unclear what you are trying to convey with that line. The same is true for "percent water," "air temperature," and so on. These are explained in the text, but it's not clear in Figure 4a.

- Figure 4b, there are numerous labels "rock glacier," "patterned fen," etc. It would be helpful if there were lead lines. In its current state, these look like a bunch of words on an image, and it's unclear what you are trying to convey.

Reply:

Figure 4a. We have elaborated on each of these in the Methods section, and on Supplementary Table 3.

Figure 4b. We have adjusted the caption to more clearly explain that the position of the words indicate landform loading scores for the RDA.

Reviewer Reply Summary

We have provided detailed replies to all reviewer comments, most of which have helped to clarify the Manuscript, resulting in minor revisions to the text. We have acknowledged the contributions of the reviews by Dr. Allard and Dr. D. K. Swanson, and of one anonymous Reviewer, that have improved the Manuscript.

The title has been shortened to 15 words as per Dr. Alves request.

" Permafrost landsystems define regional variability in climate change effects on northern environments"

REVIEWER COMMENTS

Reviewer #1 (Remarks to the Author):

Comments on the revised manuscript " Permafrost landsystems define regional variability in climate change effects on northern environments"

First, I notice that the title was changed. The original title was "Permafrost landsystems provide the key to understanding climate change effects on terrain and ecosystems". The paper has now a heavier focus on regional variability of both the permafrost landform patterns and, as a consequence, of the climate change impacts on permafrost thawing. As mentioned in the text "the emphasis is that systematic and integrated permafrost mapping provides a critical vehicle for generating new understanding of regional variability of permafrost characteristics". I interpret that such understanding of regional variability is currently inadequately represented by general models (mostly circumpolar in scale) based on spatial climate models and other datasets of terrain variables. I concur with the authors that permafrost landforms being good indicators of permafrost properties, and also the heritage of past climate and ecological changes, they must necessarily be a strong component of the capability for predicting permafrost conditions across landscapes and for predicting the reaction of permafrost to climate change. This is what make this paper important for the research community and the public.

The authors now state clear objectives at the beginning of the paper. They have added a table (No 2) in the supplementary information showing the existing land classification framework of the NWT, which helps very much to understand the context of application of their proposed landsystem approach. I understand that they consider the level IV ecoregions of this framework as corresponding to the permafrost land systems (this is also stated in their rebuttal letter), although I sense that the text does not seem to be so unequivocal on the matter at some places.

The statistical analysis (RDA) is well explained and the addition of figure 6 showing the loadings of the landforms and of their assemblages across the level IV ecoregions improves very much the communication of the key messages of the paper.

My second reading of the Manuscript raised a question in my mind: I think the authors must explain by what scientific process they identified the 28 landform types that are the basic mosaic pieces of landform assemblages that reoccur in the permafrost landscape. They have acquired a

unique knowledge of the variety of permafrost landforms in NWT and were finally able to classify them into this set of 28. One or two sentences reporting on their experience would be appreciated. This is important because the variability of occurrence of permafrost landforms in the region reflects the variety of permafrost conditions. These key landforms are also the basic elements of their RDA analysis.

I am adding an annotated manuscript with a few commentaries (mostly suggestions for small changes) for the authors to consider.

Michel Allard

Response

We appreciate Dr. Allard's insightful comments on our paper and his positive assessment of the revised Manuscript, in particular, his observation that permafrost landforms reflect permafrost properties and the heritage of past climate and ecological changes. As such, integrated landform mapping must be a fundamental component of predicting regional variation in permafrost conditions and climate change impacts on landscapes, making the analyses and concepts important to the research community and the public.

We appreciate that the Reviewer finds the objectives clear and that Figure 6 contributes positively to the paper's overall message.

In response to Dr. Allard's comment, we state early in the paper that level IV ecoregions are conceptually similar to "Landsystems". Furthermore, we provide 2 additional sentences explaining how the 28 permafrost landform types comprising the data analyzed in this paper were determined. We also point to a published research paper on the development of this unique dataset, to Figures 1 & 2, and to Supplementary Table 1. These additions make clear that the 28 landforms in the inventory reflect the diversity of permafrost conditions across the NWT (and most permafrost regions), as emphasized through edits to the last two paragraphs of the Introduction. We also appreciate Dr. Allard's minor comments in the annotated Manuscript and have made the majority of the suggested small changes, as described below.

Reviewer Replies

P3. L87-89. but it is obtainable with more field surveys and characterization of representative sites. This is necessary in particular for construction, infrastructure and land management. Maybe say "difficult to predict at regional scales..."

Response: Minor editorial suggestion was implemented.

P4. L98. "regional" is confusing here because some levels of ecoregions are defined by climate expressed by vegetation. Maybe say "spatial"

Response: Minor editorial suggestion was implemented.

P4. L105-109. note here that many of the permafrost landforms have developed or are imprinted over various Quaternary landforms and surficial deposits. This is important because permafrost ground ice content is often associated with sediment composition and fabric.

Response: We appreciate the comment and implemented the editorial modification to emphasize the Reviewer's point.

P4. L110-11. I suggest: are linked to the different permafrost characteristics across...

Response: We have addressed the comment through a minor editorial revision.

P4. L112. here make ref. to supp. table 2 (level IV ecoregions)

Response: Reference to Supplementary Table 2 has been added.

P4. L115. on continental scale, the gradient is mostly climatic

Response: We appreciate the comment and indicate we sample across a "continental-scale ecoclimate gradient"

P5. 124. you should change paragraph here, you are now taking the reader to the RDA.

Response: We did not break the paragraph here as it doesn't seem appropriate.

P5. L126. see comment above. But you must explain how, by what method, you came to the recognition that these 28 permafrost landform types are in fact all of the basic combinations of surficial deposits, ground ice contents, permafrost local landforms, organic cover and contents and vegetation that compose the various permafrost landscapes of NWT (your study area).

Response: We appreciate the comment. We have adjusted the text to mention the dataset used, reference a paper that describes the mapping methods in detail, and include reference Supplementary Table 1 (which lists all landforms and the permafrost conditions they indicate). We also point to Figures 1 & 2, which are schematics and an associated narrative of how landforms evolve and reflect surficial deposits, ground ice, soils, and vegetation combinations. We also mention here the benefits and constraints of the method, which utilized Sentinel 2 imagery with a 10m pixel resolution.

P5. L135. you already mentionned in 14 level IV ecoregions above

Response: We appreciate the comment.

P5. L140-145. True. Also an original vision using the landforms as key indicators of permafrost conditions. I concur 100%. However the best integrated level of permafrost conditions would be at the "permafrost land type" level, i.e. at a larger mapping scale. In fact, rather than mapping the land types, you have identified the 28 key landforms that are recurrent across the landscape.

Your statistical analysis bears on the spatial occurrences of these key landforms in the sampling grid allows a regional appraisal of the variability of permafrost conditions. This is OK. However my feeling is that the reader has too much effort to do to understand this simple approach because of complex text and some "overtheorizing". Your choice of communication approach.

Response: We appreciate the comment and are comfortable with our communication approach. We don't feel it is over-theoretical.

P6. L153-165. nice figure and excellent explanations in the caption.

Response: Thank you.

P8. L194-213 great concept. Some confusion remains. No 3 shows the organization of a partial (block diagram) permafrost land system (assemblage defines the concept of landsystem). Now, does No 4 show a mosaic of permafrost land forms or a mosaic of permafrost land systems (shades of blue-grey blocks) that compose an ecoregion level IV (here Tuktoyaktuk coastal plain) or are the grey-blue blocks landform types forming an assemblage within the Tuktoyaktuk coastal plain which is the Land system). Sorry, I feel being not bright.

Response: We appreciate the need for clarity. The patterns in #4 show distinct landform assemblages and their spatial distribution across the level IV ecoregion (landsystem). We clarify this through a minor modification to the figure and caption text.

Text on Figure 3.

"#4. Permafrost landsystems

Complex mapping unit composed of permafrost landform assemblages describing patterns and variation within the landsystem."

P9. L236-238. here I think you should say what axes 1 and 2 correspond to (1-climate gradient expressed by some variables, 2- topographic relief expressed mostly by mass wasting landforms). This would help the reader get into the following text.

Response: We provide a sentence to briefly summarize what the site scores along the 2 axes reflect to help a reader follow the RDA results.

P9L239. what is the scientific hypothesis that underlies this anticipation?

P9L239. correspond to ?

P9.L239-241. confusing here. are land systems of regional-local mapping scale or continental? or do you mean that they are recurrent across the landscape at continental scale?

P9.2239-241. land systems ?

Response: The Reviewer raises several minor queries regarding this topic sentence. We appreciate Dr. Allard's comments and have restructured the sentence with a minor revision to address these points, focusing mainly on clarifying that the RDA plot, stratified by ecozone/ecoclimate zone, captures broadscale variation rather than "landsystem" scale variation. We don't adjust language here to mention "landsystems". We do not state a hypothesis here; rather, we summarize the results.

" Permafrost landform assemblages capture a continental-scale gradient in permafrost conditions, and additionally, reflect the significant variation in substrate, ground ice, and ground temperature conditions that characterize heterogeneity of this broad ecoclimate transition indicated by the site scores along RDA axis 1 (Fig. 4a). "

P10.L280-282. ? how do you quantify surficial geology and make it a variable in RDA? there is no value for surficial geology in Supplementary Table 3. Is permafrost probability also a quantitative value like temperature and precipitation ?? does that probability derive from a model or field checks?

Response: In testing, surficial geology was included as a categorical variable (presence/absence of the dominant class and the total number of different units). Modeled permafrost probability was extracted from Obu et al. (2019) and used as a continuous variable in testing.

P10. L281. why not observed (sample sites) ground ice abundance? as drilled in figure 3.

Response: The explanatory variables in the RDA are restricted to broad-scale published products for which data can be extracted for the inventoried grid cells.

P10. L282 In Obu et al. permafrost probability in % is based on modelled surface temperatures. One can doubt this variable adds more explanatory power to the RDA which already includes all key climate parameters.

Response: We appreciate the comment, and we agree, which is why we did not include it in the final model.

P10. L282-288. agree. the map of landsystems would provide a great sampling base for validating permafrost properties across the region through characterization of representative sites. The predictive value would be high with a minimum (optimal?) number of sampled field sites.

Response: We appreciate the comment, and the point is elaborated on in the discussion.

P13. L363. Here, do you mean Holocene climate history and (or) recent climate evolution? the following lines suggest it is mostly past climate changes.

Response: Minor editorial adjustment clarifies that we mean Holocene climate history.

P14. L391-393. what level of ecological region region? do you mean High landform diversity within some ecotonal level-IV ecoregions?

Response: We appreciate the comment, and a minor editorial modification was implemented.

P14. L396-398. This repeats what you said a few lines above

Response: We appreciate the comment – the minor editorial modification above distinguishes the first sentence from the last in this paragraph.

P15. finally landsystems and level IV ecoregions are the same? (you say it also in your rebuttal letter): same definition, same mapping scale? I feel that either you state clearly that the already mapped land regions level IV are the assemblages of landforms or you propose to change their designation for "permafrost land systems."

Response: We appreciate the comment. We indicate early in the paper that Level IV ecoregions are conceptually similar to landsystems. We are utilizing this existing land classification framework to demonstrate the applicability of permafrost landsystem concepts, and also a scalable mapping approach for deriving permafrost landsystems in a data-driven empirical framework. We have expressed this earlier in the Manuscript – in the Introduction, and again in the results.

It is beyond the scope of this paper to suggest that a particular jurisdiction adjust terminology.

Pg 16 F6 maybe this graph would be more intuitive to read if the low % values had a light tone (colour) and the high % values a darker tone (color)

Response: We used the viridis color scale, which was developed for data visualization and performs well in monochrome and across various types of colorblindness. The light/yellow end of the spectrum is commonly used to represent higher values.

Figure 6 caption ?? why is this here?

Response: We make a minor editorial adjustment to clarify that: "The Ecoregion labels indicate Level IV ecoregions Level II & III and the Level II & III ecoregions they nest within are in italics."

P17. 468. not an easy sentence to read..

maybe say something like "Permafrost landsystems provide a conceptual framework for integrating known site level permafrost conditions into landform assemblages and mapping them at the regional scale"

Response: The sentence was modified to improve clarity.

P17 474-477

landsystems making the patterned organization within ecoregions (???)

Response: We have made a minor adjustment to improve clarity.

P17 483-486. is this necessary for the purpose of the paper? I think no.

Response: We agree.

P18 L500. geology and soil mapping

Response: Minor editorial modification was implemented.

Reviewer #2 (Remarks to the Author):

Comments on "Permafrost landsystem approaches provide the key to understanding regional variability in climate change effects on northern environments", Steven V. Kokelj et al. Nov. 2025 revision.

by Dr. David K. Swanson (recently retired from U.S. National Park Service, Fairbanks, Alaska)

I thank the authors for their conscientious responses to my and the other reviewers' comments. My few comments on this revised draft of this paper relate mostly to the newly added cluster analysis. I found this analysis to be very interesting and believe it adds a lot to the paper. This analysis provides a statistically-based picture of the composition of the Landform Assemblages. Below are a few specific comments.

Response: We appreciate the positive feedback from Dr. Swanson on our revised version of our Manuscript. We provided responses to specific comments below.

Page 5. The introductory section provides a brief overview of the methods that mentions only the Redundancy Analysis. I recognize that the methods overview here must be very short, but given

the success of the cluster-heatmap analysis, I think it deserves mention too.

Response: We have added a sentence in the short methods section at the end of the Introduction referencing the clustered heatmap.

Also on p. 5, this would be a good place to note that the presence/absence data for the 28 landforms by grid cell is your sample of the Permafrost Landform Assemblages. In other words, your list of the landforms present in a cell is the Permafrost Landform Assemblage for that location (given the limitations that the sampling method has, as all of them do). This would help the reader understand how the RDA sheds light on the properties of the Assemblages, because the input data to the RDA is a direct sample of the Assemblages.

Response: We have added a few words to help clarify this point.

Page 12, lines 4-5. The sentence here should read "Figure 6 is a clustered heatmap that explores the natural groupings of permafrost landforms INTO assemblages..." Figure 6 is the place where we get to see which landforms actually group together into assemblages. This info is also available in part from Figure 4, but here the reader must infer the assemblages based on proximity between labels on the diagram, and also their distribution is based only on how they relate to environmental variables (...with only $16.6\% + 3.1\% = 19.7\%$ of the variability portrayed).

Response: Thank you for the comment. We have adjusted the caption text accordingly.

Page 13, 3rd line from the bottom "more gently rising richness curve for the Richardson Plateau". At the scale of Figure 5 at least, the curves for RP and EBH seem to nearly overlap.

Response: We appreciate the comment and emphasize it to be a "slightly" gentler rise. We feel its worth pointing out.

Figure 6 is great. I would experiment with the color scheme to show better contrast for the landforms that occur at low frequency.

Response: We appreciate the comment. We have kept the color scheme for reasons addressed in a comment by Dr. Allard, and also because the resolving variation of landforms at high frequency is more important for understanding overall sensitivity of the landsystem. However, we may choose to explore different ways of representing landforms that are "environmentally important" but spatially infrequent in future publications.

Page 26, Methods - Statistical approach. The RDA methods are described in great detail, but documentation for the cluster analysis is very brief. The R package "pheatmap" is cited as the method used for the cluster analysis, but this package can be used to plot heat maps made by a variety of clustering methods. Based on the results (Figure 6) it appears that you collapsed the data into a matrix of Level IV Ecoregions (rows) and landforms (columns), with matrix values being percents of the sample cells for each Ecoregion that contained that landform. I could be wrong about this, so please describe the initial matrix that was subjected to clustering, how the distances were calculated, and what cluster method was used. The defaults in pheatmap are Euclidean distance and complete linkage clustering, which is a reasonable choice but not

necessarily optimal. You have 3 major clusters, and one of the clusters is quite large with chaining issues. I'd give Ward's method a try and see if it gives better results, if not here then in a future paper. Also, a more fundamental distance measure for the landform clustering might be simply based on the count of 7.5x7.5 km grids where each pair of landforms was both observed. I look forward to reading about this in future publications!

Response: We greatly appreciate the comment. We have provided additional information on the structure of the input matrix in the Methods section. Also we clarify that in "pheatmap" we use Euclidean distance and complete linkage clustering. We experimented with other methods, including Ward's, which provided largely similar results with some minor differences. We were unable to find indications that either method was more appropriate for the current data and we opted for the complete linkage method. We will consider this in more detail exploring permafrost landsystem clustering in the future.

Also in the Methods, you should probably also say which method you used with r function "specaccum". Presumably it was the default ("exact"), but it's best to specify this.

Response: We appreciate the comment and have updated the methods to clarify.

Reviewer #3 (Remarks to the Author):

I believe the authors did a good job of addressing reviewer comments. The clarifications made greatly improve the Manuscript and I believe this provides valuable information to assist users at better understanding permafrost conditions at a regional scale. I have not additional comments prior to publication.

Response: We appreciate the comment